# Dual-role iron species in photoelectrocatalytic radical trifluoromethylation with trifluoroacetates

Sara Fernández-García[1,3], Sara Cuadros [2,3], Irene Bosque [2] ✉,
Jose C. Gonzalez-Gomez [2] ✉ & Francisco Juliá-Hernández [1] ✉

Trifluoromethylation remains a fundamental transformation in drug discovery, underpinning many top-selling pharmaceuticals. Emerging developments in iron ligand-to-metal charge transfer (LMCT) catalysis have redefined fluoroalkylation chemistry by unlocking the repurposing of readily available trifluoroacetates as radical trifluoromethyl sources. Recent (hetero)arene trifluoromethylation methods generally require stoichiometric inorganic oxidants to turnover photoactive iron catalysts, thereby limiting their broader applicability. Herein, we present an integrated photoelectrocatalytic strategy employing in situ-generated multifunctional iron species to achieve C(sp²)–H trifluoromethylation without stoichiometric oxidants and generating only traceless byproducts. Mechanistic studies identify catalytically active iron species exhibiting a dual synergistic function: promoting the photo-decarboxylation of trifluoroacetates and mediating catalytic redox turnover. This protocol offers mild, tunable, and scalable conditions powered by visible light and electric current. It features a broad substrate scope, enabling the functionalization of diverse (hetero)cyclic scaffolds, including challenging electron-rich and easily oxidizable substrates, thus demonstrating its potential for sustainable late-stage modification in pharmaceutical synthesis.

Fluoroalkyl groups have emerged as integral structural motifs in medicinal chemistry owing to their profound impact on the physicochemical and pharmacokinetic properties of drug candidates[1,2]. The trifluoromethyl group (CF₃) still stands out as a major player in this field as its incorporation can markedly improve metabolic stability, lipophilicity, and membrane permeability, while also modulating molecular conformation and facilitating binding affinity[3]. As a result, trifluoromethylation reactions continue to play a central role in the design of therapeutic agents across a wide range of disease areas, often through the direct derivatization of C(sp²)–H bonds in the parent compounds[4]. A compelling example of this impact is the conversion of the naturally occurring nucleoside deoxyuridine into trifluridine via C(sp²)–H to C–CF₃ bond substitution, leading to a 400-fold increase in

DNA uptake[5] and establishing trifluridine as one of the top-selling small-molecule drugs with broad antiviral and anticancer applications (Fig. 1A)[6]. Beyond traditional closed-shell approaches[7,8], radical C–H trifluoromethylation has become increasingly favored due to its ability to directly functionalize lead compounds without the need for prior activation. Central to this strategy is the generation of trifluoromethyl radicals (·CF₃), which have driven the development of a wide array of tailored trifluoromethylating reagents displaying high enthalpic driven forces (Fig. 1B)[4]. Photoredox catalysis has significantly broadened the scope of these methodologies by offering an efficient platform for radical generation under mild conditions via photoinduced single-electron transfer (SET) processes[9,10]. However, the broader application of these methods may be constrained by the cost and accessibility of

[1]Facultad de Química, Centro Multidisciplinar Pleiades-Vitalis, Universidad de Murcia, Campus de Espinardo, Murcia, Spain. [2]Instituto de Síntesis Orgánica (ISO) and Departamento de Química Orgánica, Universidad de Alicante, Apdo. 99, Alicante, Spain. [3]These authors contributed equally: Sara Fernández-García, Sara Cuadros. ✉e-mail: irene.bosque@ua.es; josecarlos.gonzalez@ua.es; francisco.julia@um.es

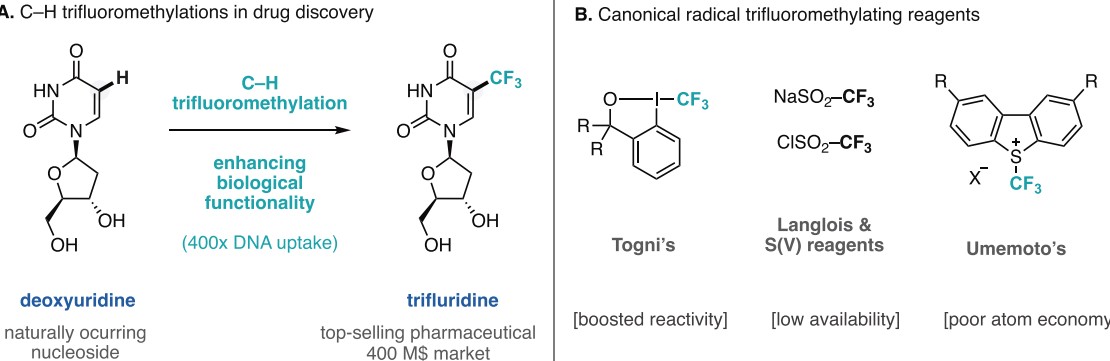

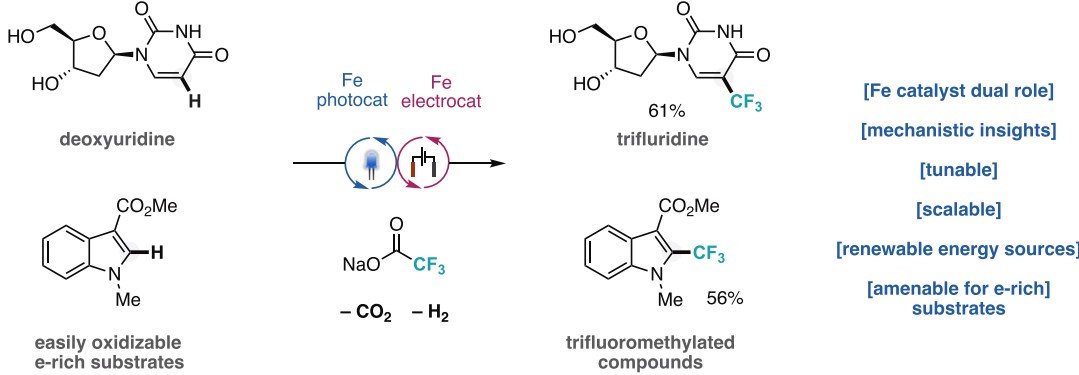

**Fig. 1 | Radical trifluoromethylation of organic compounds. A** Impact of trifluoromethyl incorporation in drug discovery and development. **B** State-of-the-art radical trifluoromethylating reagents. **C** Photodecarboxylation of trifluoroacetates with Fe LMCT catalysis[11]. **D** Radical C–H trifluoromethylation of organic molecules with trifluoroacetates by a synergistic iron electro- and photocatalysis.

specialized reagents, as well as by scalability challenges that impact their practical utility[4]. To solve this, our group has recently pioneered an alternative approach by repurposing trifluoroacetates, the most available and inexpensive chemicals containing trifluoromethyl groups, as $CF_3$ radical precursors for the trifluoromethylation of (hetero)arenes[11]. The use of trifluoroacetates as radical trifluoromethylating reagents has been limited by their high oxidation potential, $E^{ox} > +2.00$ V ($vs$ $Fc^+/Fc$)[12], often necessitating harsh reaction conditions for decarboxylation, which restricts their applicability in the functionalization of complex molecules (Fig. 1C)[12–15]. In our prior work, we reported the design of an iron-based photocatalyst that operated via an inner-sphere electron transfer mechanism[11], effectively circumventing the thermodynamic redox constraints typically associated with redox-based methodologies, including canonical photoredox catalysis[16,17]. This process entails the coordination of trifluoroacetate to an Fe(III) active species, forming a carboxylate

complex that, upon visible-light excitation to a dissociative ligand-to-metal charge transfer (LMCT) state, undergoes homolytic cleavage of the Fe−O bond[18–21]. This generates transient carboxyl radical species that allow the formation of trifluoromethyl radicals after rapid decarboxylation, by transforming a bimolecular process (outer sphere SET) into an intramolecular inner sphere SET (Fig. 1C). We capitalized this photodecarboxylation technology to the direct trifluoromethylation of C(sp²)–H bonds in a wide range of diversely functionalized unsaturated (hetero)cycles. Other research groups have subsequently adopted this strategy for further synthetic applications[22]. Complementary approaches leveraging Fe-LMCT catalysis for the fluoroalkylation of alkenes were also discovered[23–27].

Although our previous protocol exhibited excellent chemoselectivity, demonstrating broad functional group tolerance and compatibility with structurally complex molecules, its practical application and scalability were limited by the requirement for stoichiometric

amounts of an inorganic oxidant ($K_2S_2O_8$) to regenerate the active catalytic species in the context of a net oxidative transformation[11]. Accordingly, we anticipate that this reactivity concept could exert a broader impact in the field of trifluoromethylation if the turnover step is achieved via a cleaner and more streamlined approach, thereby enhancing scalability and expanding its practical applicability. However, achieving this goal is far from trivial, since the Fe(II) to Fe(III) turnover event has been identified as the rate-determining step of the transformation[11]. Moreover, realizing this objective would provide a valuable opportunity to elucidate previously unexplored mechanistic features arising from the synergistic interplay between Fe-mediated LMCT photodecarboxylation catalysis and other enabling technologies. To this end, we initially hypothesized that the reaction could be boosted by implementing an electrochemical oxidation, altogether constituting a photoelectrocatalytic strategy[28,29] through the integration of electrochemical oxidation[30] and iron LMCT photodecarboxylation[31,32]. Recent photoelectrochemical protocols have demonstrated the feasibility of trifluoromethylating organic molecules using trifluoroacetic acid (TFA) or trifluoroacetate salts[33-35]. While these approaches feature elegant reaction designs, they are often limited by a narrow substrate scope, incorporating only electron-neutral or electron-deficient (hetero)arenes presumably due to undesired oxidative degradation of electron-rich substrates[33,34], or require the use of specially engineered heterogeneous electrodes[35,36]. In view of this, a practical photoelectrocatalytic protocol for the trifluoromethylation of (hetero)aromatics involving the direct decarboxylation of trifluoroacetates and that could accommodate relatively electron-rich substrates with standardized techniques remains a significant and unmet challenge. If successful, such a synthetic protocol would hold significant potential for practical applications, particularly in medicinal chemistry, by enabling the direct trifluoromethylation of C(sp$^2$)–H bonds in multiple (hetero)cyclic scaffolds through the repurposing of trifluoroacetate feedstocks and employing tunable and scalable photoelectrocatalytic settings while generating only traceless $CO_2$ and $H_2$ gaseous byproducts.

In this communication, we report the realization of this concept through a reactivity paradigm in trifluoromethylation reactions by integrating a tunable electrocatalytic strategy with photocatalytic decarboxylation of trifluoroacetates via dual-role iron catalysis (Fig. 1D). This approach enables the incorporation of $CF_3$ groups in a wide range of substrates, including oxidation-sensitive, electron-rich (hetero)arenes, employing trifluoroacetates as trifluoromethylating reagents under standardized conditions. The protocol is characterized by the in situ formation of both electrocatalytic and photocatalytic iron species integrated in a synergistic mode of action, enabling the decarboxylation of trifluoroacetates under clean, scalable, and mild conditions using visible light and electric current as renewable energy sources. Notably, this photoelectrocatalytic method expands the substrate scope to incorporate electron-rich (hetero)arenes, which remain challenging for existing photoelectrochemical methodologies[33,34].

## Results and discussion
### Mechanistic studies and hypothesis
At the outset of our investigations, it remained unclear whether the photocatalytically active iron species could be regenerated by implementing an electrochemical oxidation. To this end, we started investigating the speciation of iron complexes under the reaction conditions previously reported by our research group[11], which involved Fe(OTf)$_2$ (OTf, trifluoromethanesulfonate) as the precatalyst in combination with 4,4'-dimethoxy-2,2'-bipyridine (L1) in MeCN at room temperature (Fig. 2A). Cyclic voltammetry (CV) measurements were carried out to evaluate the feasibility of oxidizing Fe(II) intermediates through an electrochemical process, which underpins the catalyst turnover pathway. A mixture of Fe(OTf)$_2$, L1 and sodium

trifluoroacetate in a 1:1:60 molar ratio in MeCN exhibited two oxidation peaks, with the first redox couple becoming reversible only upon addition of trifluoroacetic acid (TFA) (Fig. 2B), accompanied by a slight shift in the corresponding half-wave potential ($E_{p/2}$). Furthermore, replacing TFA with $HClO_4$ produced a similar CV profile, featuring two reversible redox couples, likely due to the in situ formation of TFA (Fig. S17). These results indicate that, in addition to serving as a proton source required to balance the electrochemical reaction through hydrogen evolution, TFA also influences the stability of the species involved in the first redox pair (Fig. S19)[23]. To identify the nature of these species in solution, we recorded cyclic voltammograms of various combinations of the reaction components in the presence of TFA (Fig. 2C). A solution of Fe(OTf)$_2$ and sodium trifluoroacetate displayed an irreversible CV with $E_{p/2}$ = +0.10 V (vs. Fc$^+$/Fc) (Fig. 2C, green trace), consistent with the presence of non-ligated Fe(II) species I (Fig. 2A). Upon addition of an equimolar amount of L1 and 1.3 equivalents of TFA, the two-redox-pair system emerged (Fig. 2C, orange trace), with the first reversible wave at $E_{1/2}$ −0.10 V (vs. Fc$^+$/Fc) consistent to the formation of ligated iron trifluoroacetate species II (Fig. 2A). The effect of TFA addition may arise from the formation of TFA-bound species through hydrogen-bonding interactions, as reported in other purely photochemical Fe-mediated decarboxylations of trifluoroacetates[23]. The second reversible wave at $E_{1/2}$ + 0.36 V (vs. Fc$^+$/Fc) is attributed to the formation of a homoleptic trisbipyridine-type Fe(II) complex (III). This assignment is supported by comparison with the CV of a mixture of Fe(OTf)$_2$ and L1 in the absence of sodium trifluoroacetate and TFA (Fig. 2C, blue trace), as well as by literature data[37]. Importantly, the addition of TFA to this mixture resulted in the appearance of the two-redox-pair system, indicating that both L1 and trifluoroacetate ligands participate in the coordination sphere of iron species II. The photostability of the iron species under reaction conditions was investigated by comparing the cyclic voltammograms of Fe(II)/L1/sodium trifluoroacetate/TFA mixtures, both with and without 405 nm irradiation (Fig. 2D). While the CV of the trisbipyridine-type Fe species (III) remained unaltered upon irradiation, the redox pair corresponding to the ligated Fe trifluoroacetate species V exhibited pronounced alterations, underscoring its photochemical reactivity. We then evaluated the photoinduced decarboxylation of in situ-formed ligated (V) and non-ligated (IV) Fe(III) trifluoroacetate species via LMCT. To this end, mixtures of Fe(OTf)$_3$, NaO$_2$CCF$_3$ and TFA in acetonitrile were prepared in the presence and absence of L1, and their evolution under 405 nm irradiation was monitored by UV-visible absorption spectroscopy (see Fig. S35-36). Notably, while the photodecarboxylation at ligated Fe(III) trifluoroacetatate species (V) required 20 h to reach completion, the corresponding non-ligated Fe(III) species in the absence of L1 were consumed in only 30 minutes, indicating a faster LMCT rate for IV (see Fig. S39). Complementary UV-visible spectroscopic analysis of a Fe(II)/L1/sodium trifluoroacetate/TFA mixture under reaction conditions further supported the formation of intermediate III, as evidenced by direct comparison with the spectrum of authentic [Fe(L1)$_3$](PF$_6$)$_2$ (see Fig. S9). Finally, the formation of species I, II, and III was confirmed by ESI-HRMS of the reaction mixture, which revealed ions corresponding to trisbipyridine-type Fe complexes and ligated and non-ligated Fe trifluoroacetate species (see Fig. S34). Based on these experimental observations, we propose a mechanistic hypothesis in which an electrocatalytic cycle synergistically converges with an Fe LMCT photodecarboxylation reaction via two plausible pathways (Fig. 2E). Under reaction conditions, intermediate II is formed, which can undergo oxidation at the anode to form the corresponding Fe(III) species V. This intermediate plays a dual role: (1) upon visible-light excitation, V can promote LMCT photodecarboxylation, affording trifluoromethyl radicals and regenerating species II through coordination with sodium trifluoroacetate (Fig. 2E, pathway A); and (2) alternatively, V may act as a redox catalyst, promoting the single-electron oxidation of non-ligated Fe species I in

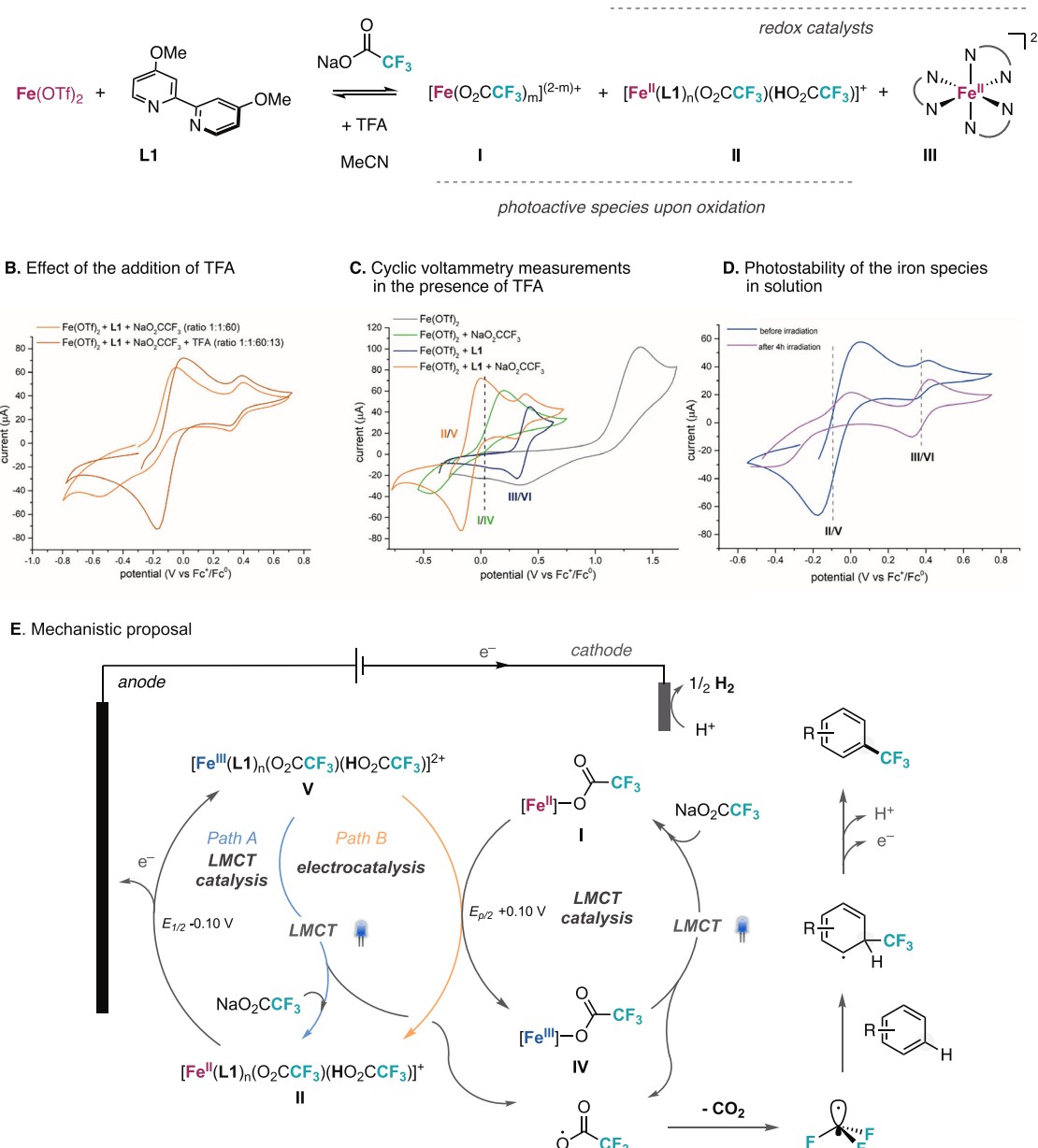

**Fig. 2 | Iron speciation and mechanistic proposal. A** Catalyst speciation by combining Fe(OTf)₂, **L1** and sodium trifluoroacetate in MeCN in the presence of TFA. **B** Effect of the addition of TFA to the cyclic voltammograms of iron species under reaction conditions. **C** Cyclic voltammetry investigation on the different iron species in MeCN solution. **D** Photostability of iron species in solutions under reaction conditions. **E** Mechanistic hypothesis integrating photocatalytic and electrocatalytic iron active species. All potentials are measured *vs.* Fc⁺/Fc.

solution, resulting in the formation of the corresponding Fe(III) intermediates (**IV**), which also promotes photodecarboxylation. It is plausible that this mediated oxidation is driven by a follow-up rapid and irreversible photodecarboxylation occurring at intermediate **IV**. This behavior markedly contrasts with a recent report on related photoelectrochemical fluoroalkylation reactions (excluding trifluoromethylation), in which Fe(II) carboxylate intermediates displayed reversible cyclic voltammograms even in the absence of ancillary ligands, illustrating the distinct speciation associated with trifluoroacetates[38]. Moreover, the necessity of TFA for stabilizing species **II** and **V** further distinguishes the $CF_3$ radical generation mechanism from our previous work, in which the photodecarboxylation of trifluoroacetates was conducted under basic conditions[11].

Finally, following $CF_3$ radical addition to the (hetero)aromatic substrate, oxidatively induced rearomatization of the resulting cyclohexadienyl-type radical could occur at the anode or be facilitated by Fe(III) trisbipyridine-type species (**VI**) in solution[39].

## Optimization studies

Guided by this mechanistic hypothesis, careful investigations to optimize the photoelectrocatalytic trifluoromethylation of C(sp²)−H bonds resulted in the conditions shown in Fig. 3. The combination of commercially available Fe(OTf)₂, **L1**, NaO₂CCF₃, and TFA resulted in the in situ formation of bifunctional iron active species. At 35 °C, 390 nm illumination, and applying a constant potential in an undivided electrochemical cell, the trifluoromethylation of the electron-rich

· Optimized conditions and control experiments: importance of the **electrocatalyst**[a]

| entry | deviation | yield of **2**[b] |
|---|---|---|
| 1 | none | 76% |
| 2 | w/o **L1** | 0% |
| 3 | w/o **L1** + [Fe(**L1**)$_3$](PF$_6$)$_2$ (3 mol%) | 41%[c] |
| 4 | w/o **L1** + **ferrocene** (10 mol%) | 90%(9%)[d] |
| 5 | w/o **L1** + **acetylferrocene** (10 mol%) | 38% |
| 6 | w/o **L1** + [(p-BrC$_6$H$_4$)$_3$N·][SbCl$_6$] (10 mol%) | 58% |
| 7 | w/o electricity | 0% |
| 8 | w/o light | 0% |

**Fig. 3 | Optimized conditions and control experiments of the photoelectrocatalytic trifluoromethylation of (hetero)arenes using trifluoroacetates, including the performance of well-defined electrocatalysts.** [a]Reactions were performed with 0.3 mmol of **1** for 24 h. [b]Yields were determined by [19]F NMR using hexafluorobenzene as internal standard. [c]7 mol% of Fe(OTf)$_2$ instead of 10 mol%. [d]Bistrifluoromethylated product. TFA trifluoroacetic acid, S solvent (MeCN), RVC reticulated vitreous carbon electrode, SS stainless steel electrode, E$_{cell}$ applied constant cell potential.

substrate **1** –with an oxidation potential much lower than that of trifluoroacetate– was successfully achieved, affording the corresponding product (**2**) in 76% yield (Fig. 3, entry 1). As discussed before, the presence of TFA served to provide the protons to close the electrochemical circuit by H$_2$ evolution at the cathode, but also to stabilize electrochemically species **II** formed in situ. No product was detected in the absence of **L1** (Fig. 3, entry 2) or even when using reaction conditions with solvated Fe(III) salts that proved successful for other fluoroalkylations different from trifluoromethylation[38] (see Fig. S33). This illustrates the key role of the bipyridine-type ligand in forming the bifunctional catalytic species for the trifluoromethylation reaction to occur. When **L1** was replaced with [Fe(**L1**)$_3$](PF$_6$)$_2$, the catalytic activity of the system was restored (Fig. 3, entry 3), suggesting the in situ formation of species **II** via dynamic equilibration in solution. Remarkably, in the absence of **L1**, the addition of ferrocene –an established redox mediator with a CV resembling that of species **II**–[40] led to the restoration of the reactivity of the system (Fig. 3, entry 4). This finding lends support to the viability of pathway B, in which species **II** facilitates the turnover of non-ligated iron species **I**, whose oxidation appears hindered at the anode. A comparable observation occurred with the addition of other well-recognized redox mediators, such as acetylferrocene[41] and magic blue (tris(4-bromophenyl)ammoniumyl hexachloroantimonate)[42] with similar electrochemical behaviors to **III** (Fig. 3, entries 5-6), enabling the trifluoromethylation reaction in the absence of **L1**. These results underscore the essential role of the electrocatalyst in enabling the turnover of Fe(II) to the photoactive Fe(III) species in pathway B, which may also contribute to facilitating the oxidative rearomatization ultimately leading to the formation of the product. Notably, we also found that the reaction proceeds effectively even at high loadings of **L1** (30–50 mol%, Table S4), conditions under which the concentration of non-ligated Fe species is expected to be minimal. This supports product formation via pathway A, where the photoactive ligated species **V** likely drives the photodecarboxylation process. Control experiments confirmed the photoelectrocatalytic nature of the protocol, as no product formation was detected in the absence of either electrical input or light irradiation (Fig. 3, entries 7-8).

## Scope of the reaction

Following optimization, the substrate scope of the photoelectrocatalytic decarboxylative trifluoromethylation protocol was investigated, resulting in the incorporation of CF$_3$ groups into a diverse range of (hetero)cyclic scaffolds (Fig. 4). The study was conducted using standardized equipment, including a Kessil lamp and an IKA ElectraSyn® device, with H$_2$ and CO$_2$ formed as the sole byproducts. A range of simple, electron-rich arenes with different substitution patterns yielded the corresponding trifluoromethylated products in good yields (**2–7**), without observing oxidative degradation of the substrates. Other substituted 6-membered heterocycles with varying electronic properties, including also electron-deficient scaffolds, were also functionalized to give the trifluoromethylated 2-pyridinone (**8**), uracil (**9**), pyrimidine (**10**), pyridine (**11**), and α-pyrone (**12**) derivatives. The reaction was also applicable to industrially relevant coumarin derivatives, with regioselective trifluoromethylation at the pyrone ring (**13,14**). Substrates incorporating 5-membered heteroarenes such as thiophene and pyrrole were also trifluoromethylated in synthetically useful yields (**15** and **16**, respectively). In substrates containing multiple reactive sites, difunctionalized products were obtained, consistent with other radical trifluoromethylation reactions[10,11]. Strikingly, our reaction design enabled the functionalization of electron-rich pyrroles and indoles (**16-18**), characterized by remarkably low oxidation potentials, through the reduction of the applied cell potential. This allowed the selective functionalization of easily oxidizable heteroarenes without decomposition, demonstrating the tunability of our method in addressing substrates that are challenging under alternative protocols[33,34].

To further highlight the practicality of our protocol, a variety of relatively complex molecules, including natural products and drug-like compounds, were submitted to the optimized reaction conditions. The methodology is efficient for the derivatization of caffeine (**19**), and doxofylline (**20**) featuring a dioxolane ring. It is worth remarking that we were able to prepare trifluridine (**21**), one of the largest-selling small-molecule pharmaceuticals[6], in a single step from readily available 2-deoxyuridine in 61% yield. The method could also be applied to the direct trifluoromethylation of the muscle relaxant metaxalone

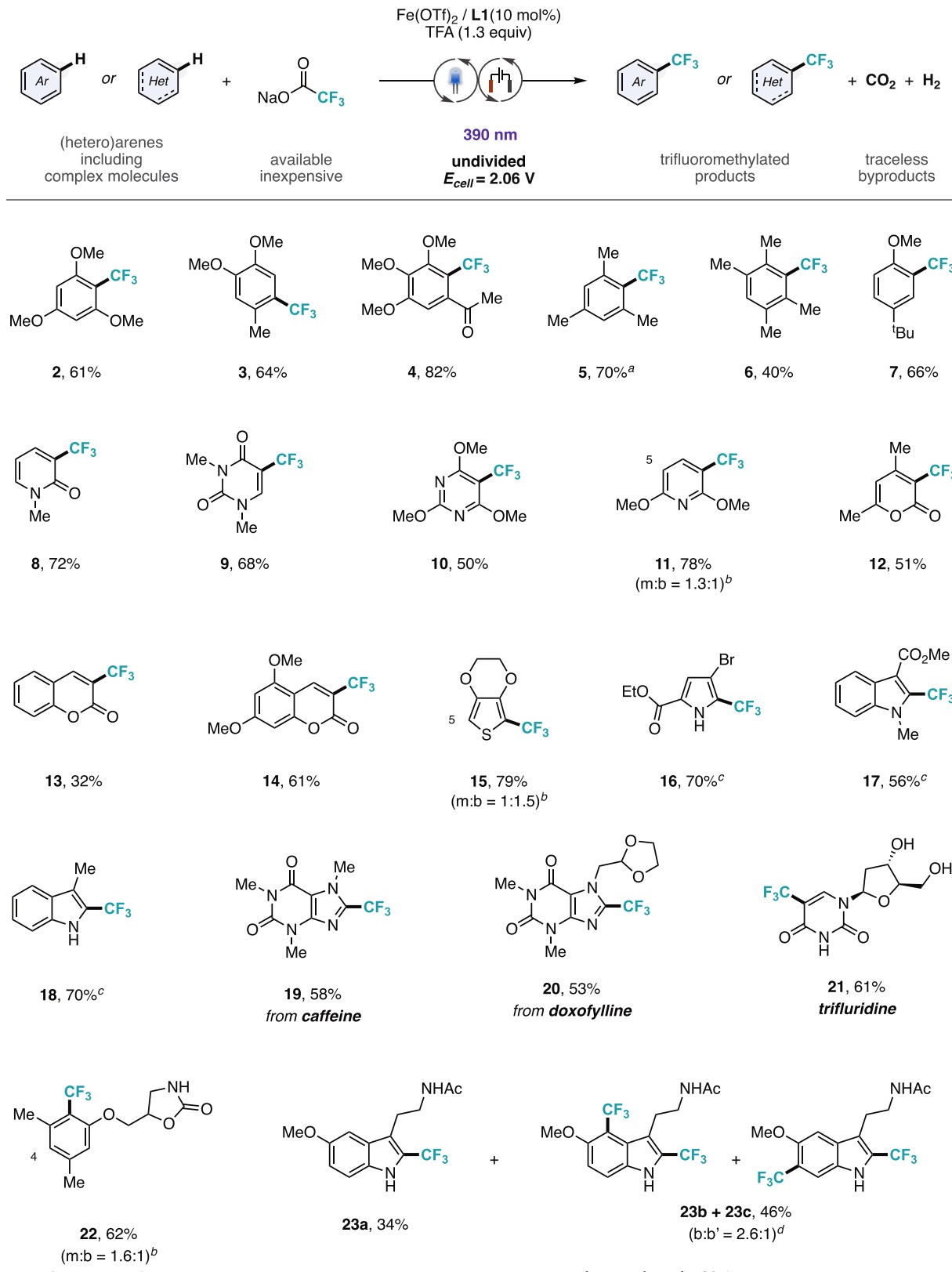

**Fig. 4 | Scope of the photoelectrocatalytic trifluoromethylation of C(sp²)−H bonds with trifluoroacetates.** Conditions: Substrate (0.5 mmol), NaO₂CCF₃ (6 equiv.), Fe(OTf)₂ (10 mol%), **L1** (10 mol%), trifluoroacetic acid (1.3 equiv.), acetonitrile (0.1 M), 390 nm irradiation, RVC/SS undivided cell constant voltage 2.06 V, 35 °C, 24–48 h. Isolated yield. *Ac*, acetyl. *ᵃ*Yield determined by ¹⁹F NMR using hexafluorobenzene as internal standard due to volatility of the product. *ᵇ*Ratio between mono (m) and bis-trifluoromethylated (b) products. *ᶜ*Using a cell constant voltage of 1.70 V instead of 2.06 V. *ᵈ*Ratio bis-trifluoromethylated isomers. Minor regioisomeric position labeled with atom number.

· Scaled-up photoelectrocatalytic trifluoromethylation

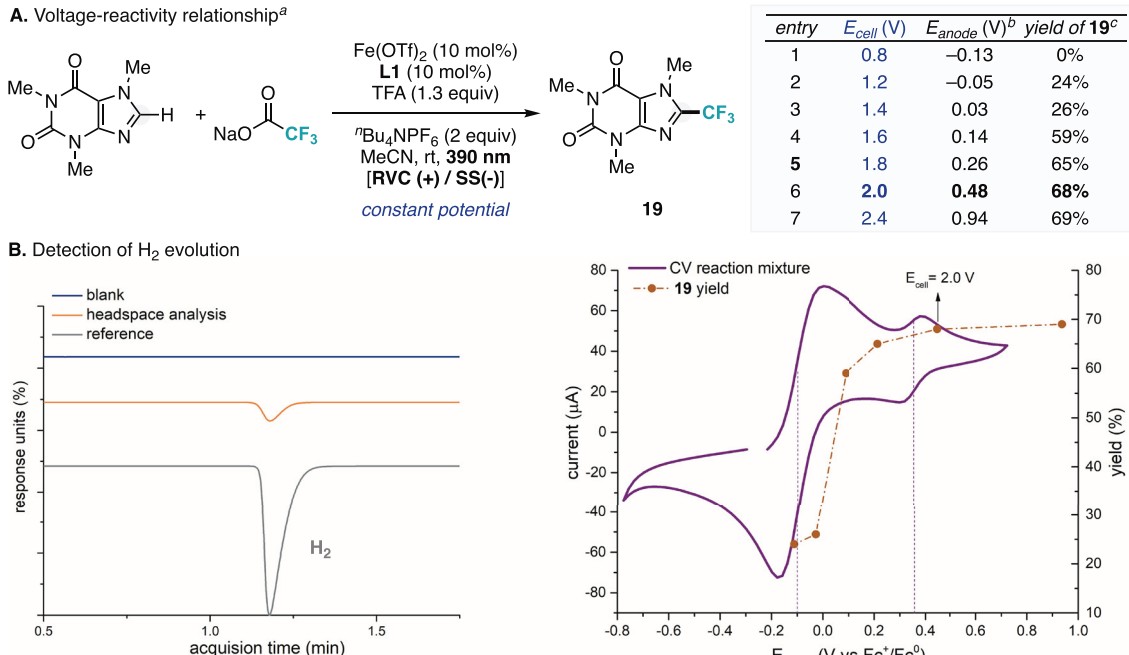

**Fig. 5 | Large-scale reactions.** Scaled-up synthesis of trifluoromethylated caffeine (**19**) and 1,3,5-trimethoxybenzene (**2**). Conditions: Substrate (5 mmol), $NaO_2CCF_3$ (6 equiv.), $Fe(OTf)_2$ (10 mol%), **L1** (10 mol%), $TBAPF_6$ (2 equiv.), trifluoroacetic acid (1.3 equiv.), acetonitrile (0.1 M), 405 nm irradiation, RVC/SS undivided cell constant voltage 2.06 V, 35 °C, 70 h. Isolated yield.

**Fig. 6 | Further mechanistic experiments. A** Voltage-reactivity relationship. [a]Conditions: **1** (0.3 mmol), $NaO_2CCF_3$ (6 equiv.), $Fe(OTf)_2$ (10 mol%), **L1** (10 mol%), trifluoroacetic acid (1.3 equiv.), acetonitrile (0.1 M), 405 nm irradiation, RVC/SS undivided cell constant voltage, 35 °C, 24 h. [b]Measured vs. Fc⁺/Fc. [c]Yields were determined by ¹⁹F NMR using hexafluorobenzene as internal standard. **B** Detection of hydrogen gas evolution.

(Skelaxin®) (**22**) and the electron-rich natural product melatonin (**23**), displaying an excellent functional group tolerance including unprotected alcohols (**21**) and carbamates (**22**). Trifluoromethylation preferentially occurred at the most electron-rich C(sp²)–H sites, likely due to polarity matching with the electrophilic character of the $CF_3$ radical[43–45]. This offers a strategy to stabilize labile sites in bioactive molecules, potentially enhancing their activity[46].

From a synthetic standpoint, the method is readily scalable through routine enlargement of the electrochemical cell. The trifluoromethylation of caffeine and 1,3,5-trimethoxybenzene was successfully performed on a larger scale without a significant decrease in reactivity. (Fig. 5).

**Further mechanistic interrogation**
Lastly, we decided to deepen our mechanistic understanding of the photoelectrocatalytic trifluoromethylation reaction. For this purpose, we evaluated the reactivity of the photoelectrocatalytic system using different applied anode potentials using caffeine as substrate (Fig. 6A). We determined the anode potential from the applied cell potential

using a third reference electrode. Interestingly, while good levels of reactivity were determined when applying anode potentials which are in the range of the CV of the trisbipyridine-type Fe species redox pair (Fig. 6A, entries 4-7), the yield of the trifluoromethylated product dramatically dropped when the mixture was subjected to lower anode potentials (Fig. 6A, entries 1–3). Although speculative, this observation aligns with the possibility that Fe(III) trisbipyridine-type species could facilitate the oxidative aromatization, thereby promoting product formation. Finally, we studied the formation of $H_2$ via proton reduction at the cathode, with TFA serving as the proton source. Hydrogen evolution was detected by headspace gas chromatography (HS-GC), occurring simultaneously with product formation (Fig. 6B).

In summary, we have developed a practical and scalable photoelectrocatalytic protocol for the trifluoromethylation of a wide range of (hetero)cyclic scaffolds, including the late-stage functionalization of industrially relevant compounds. This method employs readily available trifluoroacetate feedstocks and leverages in situ formed iron species that function dually as photocatalysts and electrocatalysts, as evidenced by experimental, electrochemical, and spectroscopic data.

The tunability of reaction conditions, mediated by catalytic redox-active species, enables the functionalization of electron-rich, easily oxidizable substrates that pose challenges for alternative methodologies. The broad substrate scope, accessible trifluoromethyl sources, and use of Earth-abundant iron active species powered by visible light and electricity collectively underscore the potential of this approach for sustainable applications in pharmaceutical synthesis.

## Methods

### General procedure for the photoelectrocatalytic trifluoromethylation of C(sp$^2$)–H bonds with trifluoroacetates

A 10 mL ElectraSyn vial equipped with a stirring bar was charged with the corresponding substrate (0.5 mmol, 1 equiv), NaO$_2$CCF$_3$ (408 mg, 3.0 mmol, 6 equiv), Fe(OTf)$_2$ (17.8 mg, 0.05 mmol, 10 mol%), 4,4'-dimethoxy-2,2'-bipyridine **L1** (10.6 mg, 0.05 mmol, 10 mol%), and TBAPF$_6$ (387 mg, 1.0 mmol, 2 equiv.). The electrodes RVC ( + )/SS (-) were then inserted, and the vial was closed with an IKA screw-cap. The hole of the screw-cap was closed with a septum, and the vial was evacuated and backfilled with nitrogen with the aid of a needle, and this procedure was repeated three times. Against a positive N$_2$ flow, dry MeCN (5 mL) and 50 μL of TFA were added via a syringe. Additional degassing with N$_2$ flow for 3 min was carried out in the reaction mixture. The reaction was stirred (650 rpm) under potentiostatic conditions (*i.e.* constant voltage = 2.06 V) using ElectraSyn 2.0 while irradiated with a 390 nm Kessil LED, at 3 cm, and cooled down using a fan. After 24–48 h, the electrodes were rinsed with Et$_2$O or EtOAc (10 mL), the reaction was quenched with saturated aqueous NaHCO$_3$ solution (2 mL) and transferred to a separating funnel. The two phases were separated, and the organic layer was washed with a saturated aqueous NaHCO$_3$ solution (2 × 15 mL), then with brine (15 mL) and dried over anhydrous Na$_2$SO$_4$. After removal of the solvent under reduced pressure, the crude mixture was purified by flash column chromatography on a silica gel column to afford the trifluoromethylated product. For products that are partially soluble in water, the reaction was quenched with Na$_2$CO$_3$ (54 mg), and the crude mixture was filtered through celite. After solvent removal under reduced pressure, the mixture was directly purified by flash column chromatography.

### General procedure for scale-up trifluoromethylation reactions

An oven-dried 100 mL Schlenk tube was equipped with a stirring bar and charged with the corresponding (hetero)arene (5 mmol, 1 equiv), NaO$_2$CCF$_3$ (4.1 g, 30 mmol, 6 equiv), Fe(OTf)$_2$ (178 mg, 0.5 mmol, 10 mol%), 4,4'-dimethoxy-2,2'-bipyridine (**L1**, 106 mg, 0.5 mmol, 10 mol %), and TBAPF$_6$ (3.87 g, 10 mmol, 2 equiv). All reagents were added under open-air conditions. The reticulated vitreous carbon (RVC) anode (50 × 25 × 3 mm) and stainless steel (SS) cathode (50 × 25 × 1.7 mm) were then inserted, and the vial was sealed with a perforated septum. The Schlenk tube was evacuated and back-filled with argon three times. Against a positive argon flow, degassed dry MeCN (50 mL) was added via syringe. The reaction was stirred (650 rpm) under constant potential electrolysis (CPE, E$_{cell}$ = 2.06 V), while irradiated with 405 nm light (EvoluChem LEDs from HepatoChem®) at a distance of 3 cm. After 70 hours, the electrodes were rinsed with EtOAc (10 mL), and the crude mixture was transferred to a 250 mL separatory funnel. The organic phase was washed with a saturated NaHCO$_3$ solution (30 mL). After phase separation, the aqueous layer was extracted twice with EtOAc (20 mL). The combined organic layers were dried over anhydrous Na$_2$SO$_4$ and evaporated under reduced pressure. The crude product was purified by automated flash column chromatography on silica gel.

## Data availability

The authors declare that the data supporting the findings of this study are available within the paper and its Supplementary Information files.

Raw data files are available in the public repository *Zenodo* at https://doi.org/10.5281/zenodo.17939074. All data are also available from the corresponding author upon request.

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

## Acknowledgements

We are grateful for financial support provided by grant CNS2023-145163 to F.J.H. funded by MICIU/AEI/10.13039/501100011033 and by the "European Union NextGenerationEU/PRTR". We also thank funding by MICIU/AEI/10.13039/501100011033, by "ERDF A way of making Europe" and by the "European Union" (PID2023-15452NB-I00 and RYC2018-024643-I to F.J.-H.; and PRE2021-099616 to S.F.-G.). Generalitat Valenciana (CIAICO/2022/017 to J.C.G.-G. and SEJIGENT/2021/005 to I.B.) and MCIN/AEI/ 10.13039/501100011033 by the "European Union NextGenerationEU/PRTR" (CNS2022-135161 to I.B.) are also acknowledged. We also thank the research support area ACTI (Universidad de Murcia) that helped with the characterization of the compounds.

## Author contributions

J.C.G.-G., I.B., and F.J.-H. conceived the project. S.F.-G., S.C., J.C.G.-G., I.B., and F.J.-H. designed the experiments. S.F.-G. and S.C. conducted the experiments, collected the data, and interpreted the results. J.C.G.-G., I.B., and F.J.-H. secured the funding and supervised the project. F.J.-H. wrote the manuscript with input from all authors.

## Competing interests

The authors declare no competing interests.
