## [Transparent Peer Review file · Nature Communications]

Dual-Role Iron Species in Photoelectrocatalytic Radical Trifluoromethylation with Trifluoroacetates

Corresponding Author: Dr Francisco Julia-Hernandez

Version 0:

Reviewer comments:

Reviewer #1

(Remarks to the Author)

The manuscript entitled "Dual-Role Iron Species in Photoelectrocatalytic Radical Trifluoromethylation with Trifluoroacetates" reports a photoelectrocatalytic protocol for C(sp²)-H trifluoromethylation using readily available trifluoroacetates as CF₃ sources. The authors propose that in situ-generated Fe species play a "dual role" as both photo- and electrocatalysts, enabling decarboxylation and catalyst turnover without stoichiometric oxidants. In my view, this work represents an incremental rather than a conceptual advance over existing Fe-LMCT and electrophotocatalytic trifluoromethylation chemistry, and therefore does not meet the novelty and impact criteria for Nature Communications.

1. The combination of Fe-based LMCT photochemistry with electrochemical oxidation has already been demonstrated in several recent studies (e.g., Campbell et al., *Science* 2024, 383, 279; Chen et al., *Science* 2024, 384, 670; Qi et al., *J. Am. Chem. Soc.* 2023, 145, 24965). The current work does not introduce a clearly distinct mechanistic paradigm or reactivity principle beyond these precedents.

2. The authors' previous report (*Angew. Chem. Int. Ed.* 2024, e202311984) already established Fe-LMCT photodecarboxylation of trifluoroacetates. The present study essentially replaces the chemical oxidant (K₂S₂O₈) with anodic oxidation to complete the catalytic cycle. While this modification improves atom economy, it constitutes an expected optimization rather than a conceptual breakthrough. Moreover, Motornov et al. (*Angew. Chem. Int. Ed.* 2025, 64, e202504143) have employed a similar iron catalyst in photoelectrochemical catalysis with a broader substrate scope.

3. The mechanistic analysis is limited to cyclic voltammetry and UV-vis spectroscopy. No direct or quantitative evidence supports the coexistence of distinct Fe species responsible for photo- and electrocatalytic functions. Operando or time-resolved studies would be necessary to substantiate the proposed dual-catalyst mechanism.

4. The substrate scope largely overlaps with known photoredox and electrophotocatalytic CF₃-introduction protocols. The functionalization of a few drug-like molecules (e.g., caffeine, melatonin) is insufficient to raise the overall impact to the level expected for Nature Communications.

Reviewer #2

(Remarks to the Author)

In this work, Francisco Juliá-Hernández et al. report a very elegant strategy for the photoelectrocatalytic radical trifluoromethylation of arenes using easily accessible alkali trifluoroacetate salts and TFA. This approach is highly powerful and could have a significant impact on the pharmaceutical and agrochemical industries by providing a more sustainable route to fluorinated molecules. This study builds upon the authors' 2024 *Angewandte Chemie* paper, in which they used a stoichiometric oxidant for a Minisci-type trifluoromethylation reaction. Clearly, the present work represents a greener approach, as it solves a long-standing problem in nucleophile cross-coupling: the use of stoichiometric amounts of reactive and aggressive oxidants, which often limit functional group tolerance and the overall reaction scope.

The authors provide an elegant solution to this issue by employing dual iron electrocatalysis and photocatalysis via a LMCT (ligand-to-metal charge transfer). They found that a "ligand-free: iron trifluoroacetate species acts as a photocatalyst, generating the trifluoromethyl radical via light-induced charge-transfer homolysis and decarboxylation. Meanwhile, tris(bipyridine)-coordinated iron serves as an electron mediator in the electrocatalytic step.

The authors present thorough mechanistic studies supported by UV-Vis spectroscopy, cyclic voltammetry (CV), and chronoamperometry, as well as hydrogen detection, all of which are discussed in detail in the manuscript and the Supporting

Information (SI). The SI is of good quality and complete.

I strongly recommend the publication of this article, as it will be an excellent contribution for the readers of Nature Communications.

Minor comments and suggestions:

- In Figure E6c, please correct the spelling of "voltajes" (should be "voltages").
- I would also like the authors to comment and discuss how the pyridyl radical, generated after the addition of the trifluoromethyl radical, undergoes oxidation, rearomatization, and proton release. Does this process occur directly at the electrode surface, or is the iron(III)L3 complex again the oxidant in this step?
- Additionally, I suggest that the authors discuss the reaction's efficiency at different ratios of the iron; L is used, as this will naturally affect the balance between "ligand-free" iron and tris(bipyridine) iron species.
- Finally, what is the effect of the bipyridine ligand structure (e.g., tert-butyl, Me or unsubstituted bipy) on the electron mediator properties?

Reviewer #3

(Remarks to the Author)

In this communication, Fernández-García et. al. report a photoelectrocatalytic methodology for the decarboxylation of cheap and abundant TFA using Fe catalysis, producing trifluoromethyl radicals that may be harnessed for the trifluoromethylation of (hetero)arenes. This work is a continuation of their 2024 ACIE manuscript (Angew. Chem. Int. Ed. 2024, 63, e202311984; Ref. 11 in the current work) in which the main advance is moving away from a chemical oxidant (K₂S₂O₈) toward electrochemical oxidation. Thus, this is the most distinguishing point of the paper. It certainly is not a big leap to do so but there are a few notable points. While photoelectrocatalysis has previously been demonstrated in Fe-mediated fluoroalkylation from fluoroalkyl carboxylates (Ackermann, Angew. Chem. Int. Ed. 2025, 64, e202504143, ref. 40 in the current work), the current work is distinguished by perhaps broader applicability towards electron-rich functionalities, as well as a potentially unique reaction mechanism. The demonstrated reaction scope appears nearly identical to the author's prior work (ref. 11), although the tolerance of electron-rich functionality (substrate 2) is notable under electrochemical conditions where anodic overoxidation is a possibility. In light of the above points, I recommend publication after addressing the following comments, which mainly concern 1) clarification of the reaction mechanism and 2) differentiation of the methodology from prior work.

1. The redox features in the cyclic voltammograms are reported vs a Ag/AgCl reference. The authors are encouraged to report the potentials vs ferrocene or SCE instead, as these values are more utilized and easily comparable in non-aqueous systems.
2. The authors state: "whereas the redox signals corresponding to complex II remained stable, displaying a higher anodic than cathodic current, suggestive of electrocatalytic behavior." In reference to Figure S20. Upon inspection of Figure S20, it appears the CV was slightly misinterpreted. Peak anodic and peak cathodic currents should not be read from absolute value on the y-axis. Please do some reading in the most rudimentary electrochemical textbooks to properly interpret the CVs. When reading from the absolute values, the peak anodic current of the last scan (dark orange line) is 25 μ A and the peak cathodic current is -11 μ A, which is perhaps why the authors made the above statement. However, the baseline in the forward wave is about 5 mA while that of the reverse wave is ca. 9 mA, meaning that the peak anodic current is 20 μ A and cathodic current -20 μ A when read from these values, indicating a fully reversible wave, as expected for species II. Absolutely no evidence for electrocatalysis can be gleaned from this CV behavior. These are the types of sloppy mistakes this field is prone to ... electrochemistry is a field unto itself. It needs to be done properly even if this is an "organic" manuscript.
3. The assignment of species III in the CVs is ambiguous. Its irreversibility and similarity to a CV of Fe(OTf)₂ + NaO₂CCF₃ suggest it is a TFA ligated Fe species, but a few questions remain. Figure 2B should include concentrations of analytes in the figure caption or figure legend. If the orange trace and green trace are equimolar Fe concentrations, why does the orange trace feature a boosted III redox wave? Figure S20 shows that this wave diminishes significantly upon cycling, suggesting that the chemical step following oxidation is depleting the precursor to III overtime. Can the authors also cycle a CV of Fe(OTf)₂ + NaO₂CCF₃ similarly to Figure S20? Does the redox wave at -0.35 V also diminish in intensity? If equimolar amounts of Fe and TFA are used, does the addition of L1 truly boost the current associated with III in the first scan, as Figure 2B suggests? This would mean that there is some Fe species in solution that contains both TFA and L1.
4. The above point is convoluted by the fact that the authors observe near quantitative formation of II via Beer-Lambert analysis, ruling out Fe species that contain both TFA and L1. If this is true, it is unclear why L1 would boost the anodic current of III on the first scan, as no IV is present in solution yet to mediate the oxidation of III to V. The claim that quantitative II is formed suggests that adding too much L1 should completely shut off reactivity, as no Fe would be left to ligate to TFA. It does not appear that the authors included an optimization of L1 concentration, unless I am mistaken. If 30 mol% L1 or greater is employed, all the Fe should be sequestered as II, and the yield should drop. Can the authors demonstrate this? Recent work published on the rxiv (<https://doi.org/10.26434/chemrxiv-2025-djp0r>) showcases TFA decarboxylation from well-defined (dien)FeTFA complexes. It is possible that the photoactive species can still contain a mixture of TFA and bpy ligands. This, of course, would be supported by the presence of product even when employing, say, 50 mol% ligand.

5. Notably, the authors demonstrate that Ackermann's conditions (ref 40) result in 0 percent yield on substrate 2, showcasing the current methodology's tolerance towards more electron-rich substrates. However, the authors attribute this to the role L1 plays in the proposed mechanism. Is it possible that L1 simply serves to decrease the required oxidation potential to access Fe(III)? Thus, the yield is 0 under Ackermann's conditions due to direct oxidation of 2 at the anode. Ackermann demonstrates successful trifluoromethylation of caffeine without a ligand, which is harder to oxidize. Under the current conditions, if L1 is excluded, can caffeine still be engaged? If not, this is likely due to the low cell potential (2.06 V), if this is raised, or constant current electrolysis is utilized, I assume caffeine can be engaged without L1. Please perform this experiment.

6. The use of mediators instead of L1 restores reactivity, including non Fe-containing mediators such as magic blue; this is a notable result. However, species III is still easier to oxidize at an electrode than these mediators, so it is unclear as to why the harder to oxidize species would serve as an electron shuttle. A separate hypothesis is that the redox mediator may help to oxidize arene radical intermediates in solution to the cation, which facilitates rearomatization after radical addition following proton loss.

7. Altogether, the mechanistic hypothesis proposed in Figure 2D is perhaps plausible, but many uncertainties remain. To me, it is clear that L1 ligates Fe(II), making it easier to oxidize to Fe(III) which is required for LMCT. The fact that no yield is observed in the absence of L1, even though no ligands are required in other work (ref. 40) is because, in this work, the potential at the anode is low. This helps facilitate reactions with electron-rich substrates. Presumably, the use of other bipyridine derivatives, or phenanthroline derivatives, can further modulate the potential required for Fe(II) oxidation. How are the CVs changed when more or similar electron rich bpy derivatives are used (R = NMe₂, tBu, Me, etc) or when more electron-poor derivatives are used (R = CF₃, CO₂Me, Cl, F, etc.)? Can reactivity be improved with a different ligand? If the required oxidation potential is moved too anodic with more electron poor bpy derivatives, does this completely shut off reactivity in the utilized potential window? Mechanistic experiments with varied bpy ligands should be carried out to provide more insight into the proposed mechanism.

8. The reaction times should be included in Figure 3.

9. Substrates 10 and 11 do not contain sterically accessible coordinating nitrogen atoms, so the claim "Notably, the catalyst remained active in the presence of substrates containing potentially coordinating nitrogen atoms" is not supported.

10. The demonstrated scope is remarkably similar to the author's previous work (ref. 11). Compare substrate 11 (this work) vs 17 (ref. 11), 8 (this work) vs 19 (ref. 11), 21 (this work) vs 33 (ref. 11), or 19 (this work) vs 29 (ref. 11). Even substrates highlighted as challenging due to their electron-rich nature (18) performed equally well, if not slightly better, in ref. 11 (substrate 24). Is there any substrate or substrate class that can be engaged in the current methodology that is not amendable to the previous conditions employed in ref. 11? Nonetheless, it is understood that elimination of a stoichiometric oxidant increases the atom economy of the transformation.

11. The authors should check their assignment of the regioisomer depicted for substrate 12, it is more likely that the radical addition occurs next to the carbonyl moiety.

12. If the redox wave for III is overlaid in Figure 6, it looks as if the yield begins to rise when accessing that couple, as opposed to the oxidation of the mediator. In fact, the yield plateaus even before reaching the E_{1/2} of Fe(II)L13 to Fe(III)L13. The yield at 0.5 V is also very comparable to the yield at 0.8 V, even though the current displayed under these two conditions are very different. This speaks to some of the remaining mechanistic ambiguity highlighted above. Please address.

Version 1:

Reviewer comments:

Reviewer #1

(Remarks to the Author)

The authors have done careful revision and detailed response. The additional mechanistic and electrochemical studies substantially improve the clarity and rigor of the manuscript, and the revised interpretation of iron speciation and CV data is appropriate. Although my view that this work represents an incremental rather than conceptual advance relative to prior Fe-LMCT and photoelectrochemical trifluoromethylation chemistry largely remains, I acknowledge that the revised manuscript more clearly differentiates this system through ligand-controlled iron speciation, the critical role of TFA, and its compatibility with electron-rich substrates. I consider my major concerns addressed and recommend acceptance.

Reviewer #2

(Remarks to the Author)

In this resubmission, Francisco Juliá-Hernández and the co-authors have thoroughly addressed all of my questions and carefully considered my suggestions. The scientific quality of the work is very high, and the results are both timely and impactful.

I have the following optional comments that can improve the clarity of the work:

1. I found the manuscript structure somewhat challenging, particularly with respect to the separation of the mechanistic discussion across the scope section. I would suggest presenting the mechanistic considerations together, either before or after the scope, to improve clarity and readability.
2. At present, the mechanistic picture involves the interplay of multiple iron species that can undergo LMCT to generate photoactive iron trifluoroacetate species capable of delivering the trifluoromethyl radical. In this context, UV-vis titration experiments could be used to quantify the relative amounts of iron chelates bearing three, two, or one ligand. Did the authors explore this possibility?
3. Importantly, reaction yields are not necessarily correlated with the rates of elementary steps. I therefore suggest examining the stoichiometric reaction kinetics (in the absence of electricity and TFA) by trapping the CF_3 radical with TEMPO, which would facilitate interpretation of the relative efficiency of the various iron species involved in LMCT. For the kinetic analysis (e.g., by ^{19}F NMR) of the stoichiometric system, the authors could employ $\text{Fe}(\text{OTf})_3$ (1 equiv, 0.05M), $\text{CF}_3\text{CO}_2\text{Na}$ (>10 equiv), and systematically vary the ligand loading (0, 1, 2 and 3 equiv), collecting several time points at low conversion (<40%). Analysis of the resulting kinetic profiles could help identify which of the non-ligated, mono-ligated, or bis-bipyridine-ligated iron species is most efficient at generating the trifluoromethyl radical.

Overall, as stated previously, I believe this work is of excellent quality, and I strongly recommend it for publication in Nature Communications.

Reviewer #3

(Remarks to the Author)

In response to reviewer comments, the authors of this work have drastically changed the mechanistic interpretation of the reaction mechanism, aligned with suggestions made in my original review of the manuscript. The effort the authors put into this mechanistic reevaluation is appreciated, and in my opinion, the mechanism is sounder than the original draft. However, certain questions remain. Chief among them is the role of Path B. Path B seems almost entirely irrelevant, and the reaction can be purely described by Path A, which is no different than many electrophotocatalytic schemes reported in the literature. I agree with Reviewer #1 that the current work is an incremental advance of the authors' prior work, with many of the defining principles of the work already having much precedent (electrochemical turnover of Fe perfluoroalkyl carboxylates demonstrated by Ackermann, substrate scope identical to authors' prior work), as I also stated in my original review. Thus, paramount to the novelty of this work is the proposed distinct mechanistic paradigm involving "dual-role" iron species. It appears to me that the authors are clinging to this mechanistic interpretation to preserve the novelty of the current work, but all signs point to a normal electrophotocatalytic pathway (i.e. Path A) that is not altogether a new concept.

In the original draft, Path A was not discussed by the authors as a possibility. Rather, all reactivity was described to an electrocatalytic turnover of I to IV (III to V in the original draft) mediated by FeL_3 . Many of my comments (example: comment #s 3, 4, 7, 12) drove after the point that reactivity could be enabled entirely by heteroleptic FeLnTFAM complexes that (i) undergo $\text{Fe(II)} \rightarrow \text{Fe(III)}$ redox at an electrode and (ii) undergo $\text{Fe(III)} \rightarrow \text{Fe(II)}$ redox via photochemical LMCT. With new experiments, the authors now include this mechanism as Pathway A in the current work. This represents a familiar LMCT electrophotocatalytic cycle, for example, identical to the scheme proposed by Nocera ($\text{Ag(I)} \rightarrow \text{Ag(II)}$ redox at an electrode and $\text{Ag(II)} \rightarrow \text{Ag(I)}$ redox via photochemical LMCT). Pathway B remains similar to the original hypothesis, but it is not sufficiently substantiated.

(1) Pathway B invokes chemical oxidation of I by V. inspection of the CV data shows that I is harder to oxidize than II, meaning the V/II couple should not be capable of inducing oxidation of I to IV. In fact, oxidation of I is ~200 mV uphill from oxidation of II, and practically zero oxidative current for IV/I exists at the $E_{1/2}$ of V/II. Given this, why do the authors argue that the IV/II couple is relevant under catalytic conditions? Furthermore, HRMS data do not even support the existence of ligandless FeTFA salts.

(2) Using the Nernst equation to predict how much IV (A^+) will be formed ... if an equivalent amount of V (Ox) is added to I (A), and II is (Red), the following equation describes the reaction equilibrium:

The ratio of $[\text{A}^+]$ to $[\text{A}]$ is at equilibrium is given by:

$$\text{Log}\left\{\frac{[\text{A}^+]}{[\text{A}]}\right\} = 8.47\Delta E$$

where ΔE is $E(\text{Ox/Red}) - E(\text{A}^+/\text{A})$. Thus, in this case, ΔE is -0.2V , and there is subsequently approximately $50\times$ more A in solution than A^+ . Given this unfavorable equilibrium implied in Path B, why do the authors maintain its importance in the overall reaction mechanism? Of course, this Nernstian treatment assumes that both the V/II and IV/I redox couples are reversible, and the latter is not. Thus, this unfavorable ET could be driven by downstream irreversible chemistry involving IV, such as LMCT. But why would pathway B, in which thermodynamically unfavorable oxidation is required, be competitive with Path A? LMCT quantum yields are also notoriously low, meaning this second irreversible step is not facile. The authors state in their rebuttal that "although intermediate V is photoactive, its evolution might be slow enough to compete with a fast

electron transfer with intermediate I in solution." Is evolution of V slow (no electrokinetics are provided), or is electron transfer to I fast (once again no kinetics). Thus this statement regarding the viability of Path B is purely speculative. V is most likely photoactive and thus can be generated at the electrode, and therefore pathway A can account for all reactivity.

(3) The authors state in their rebuttal that experimental evidence for Path B is that without L1, no product is generated. The absence of L1 would also shut down path A, which more than likely explains the lack of reactivity. The authors also show the ability for redox mediators to restore reactivity, but this is not relevant under catalytic conditions when L1 is present. It is even more irrelevant when the new ligand loading experiments are included, which conclusively show product yield even with 50% ligand loading, conditions where ligandless FeTFA is expected to be practically nonexistent. Interestingly, it appears the authors get better results when using ferrocene instead of L1 (see Fig 3). Why was this not pursued further? Ferrocene is actually cheaper than L1 as well (by a factor of 40 based off Sigma and Ambeed prices), why not use it to mediate the transformation instead of L1 when the economics and yield are strictly better?

In summary, in light of the above comments, I believe Path B should be discarded unless the reaction is run without L1 and with redox mediators. It seems likely that, under catalytic conditions, only Path A is operative. And this path is no different than standard electrophotocatalytic protocols. Path B may be operative with no ligand included, but then obviously Path A would not be functioning. The title is "Dual-Role Iron species..." but Fe would only be playing a dual role if Paths A and B were operative at the same time, and this is not necessary for product formation and also is not proven to be the case. The proposed dual-role mechanism is overcomplicated for novelty's sake, and can be reduced to simply one integrated electrophotochemical cycle (Path A).

Unfortunately, this does decrease one of the main selling points of the paper. My original recommendation was publication if the proposed reaction mechanism could be supported and if the methodology could be differentiated from prior work. Due to the "dual-role" mechanism not being in play anymore, as well as the scope not being able to be expanded beyond that of the authors' own prior work (comment #10), I leave it up to the Editor to decide if this paper should be published. Regardless, the current version of the paper should not be published unless they provide definitive proof that Pathway B is operative.

Version 2:

Reviewer comments:

Reviewer #2

(Remarks to the Author)

The authors have fully addressed my comments and suggestions; therefore, I recommend publication of the article "Dual-Role Iron Species in Photoelectrocatalytic Radical Trifluoromethylation with Trifluoroacetates" in Nature Communications as it is.

Reviewer #3

(Remarks to the Author)

The authors have performed many more experiments and while I'm still not entirely convinced about the dual-pathway, I believe they have provided data to establish that it is technically plausible. Thus the publication is suitable for publication.

“Dual-Role Iron Species in Photoelectrocatalytic Radical Trifluoromethylation with Trifluoroacetates”

(manuscript number NCOMMS-25-80469)

Point-by-point Response to the Reviewers' Comments

First, we would like to thank the reviewers for their valuable feedback and insightful questions, which have prompted us to further refine the mechanistic understanding of our transformation. In response, we have expanded our mechanistic investigations by extending the electrochemical and spectroscopic studies to include conditions in the presence of trifluoroacetic acid (TFA). We have found that, in addition to supplying protons to close the electrochemical circuit through H₂ evolution at the cathode, TFA also contributes to the electrochemical stabilization of certain iron species generated under the reaction conditions. The presence of TFA alter the shape of the cyclic voltammogram of a mixture of Fe(OTf)₂, **L1** and sodium trifluoroacetate (molar ratio 1:1:60), giving rise to a new reversible redox couple at -0.1 V (vs. Fc⁺/Fc) in addition to the signals associated with the [Fe(**L1**)₃]^{2+/3+} system previously detected under basic conditions. Based on additional CV and UV-Vis spectroscopic measurements, together with HRMS analyses, this newly observed reversible redox couple, which remains electrochemically stable over 50 CV scans, has been assigned to ligated Fe-trifluoroacetate species **II**. Further electrochemical studies conducted under illumination revealed that the redox couple **II/V** is also photoactive. These findings, along with the complementary data now incorporated into both the revised manuscript and the Supplementary Information, have enabled us to refine our original mechanistic proposal, in which an electrocatalytic cycle synergistically converges with an Fe-mediated LMCT photodecarboxylation process through two plausible pathways. Under reaction conditions, intermediate **II** is generated, which can be oxidized at the anode to form the corresponding Fe(III) species **V**. This intermediate serves a dual function: 1) upon visible-light excitation, **V** can facilitate LMCT photodecarboxylation, producing trifluoromethyl radicals and regenerating species **II** through coordination with sodium trifluoroacetate (*pathway A*); and 2) alternatively, **V** may act as a redox catalyst, promoting the single-electron oxidation of non-ligated Fe species **I** in solution, thereby generating Fe(III) intermediates (**IV**), which also contribute to photodecarboxylation (*pathway B*). Although intermediate **V** is photoactive, its evolution might be slow enough to compete with a fast electron transfer with intermediate **I** in solution. This behavior contrasts sharply with the recent study on photoelectrochemical fluoroalkylation (excluding trifluoromethylation), where Fe(II)

carboxylate intermediates showed reversible cyclic voltammograms without ancillary ligands, highlighting the unique speciation of trifluoroacetates (*Angew. Chem. Int. Ed.* **64**, e202504143 (2025)). Additionally, the need for TFA to stabilize species **II** and **V** distinguishes the CF₃-radical generation mechanism from our prior, in which trifluoroacetate photodecarboxylation occurred under basic conditions (*Angew. Chem. Int. Ed.* **63**, e202311984 (2024)).

The discussion of the updated mechanistic proposal has been added to the revised manuscript, including the following new version of Figure 2.

A. Catalyst speciation: simultaneous in situ formation of both photoactive species and redox catalysts

B. Effect of the addition of TFA

C. Cyclic voltammety measurements in the presence of TFA

D. Photostability of the iron species in solution

E. Mechanistic proposal

Experimental evidence supporting *pathway B*:

- No product formation was detected in the absence of **L1**, indicating that catalytic turnover of species **I** is not efficient at the anode.
- In the absence of **L1**, reactivity could be restored upon addition of redox catalysts with similar electrochemical properties as that of species **V**, as for example ferrocene. This indicates that species **V** may act as redox mediators facilitating **I** to **IV** oxidative turnover. This observation also highlights the ability of non-ligated Fe(III) trifluoroacetate intermediates (**IV**) to promote photodecarboxylation under acidic conditions, as previously supported by theoretical calculations (*Nat. Comm.* **15**, 6115 (2024)).

Experimental evidence supporting *pathway A*:

- Detection of a new reversible wave in the cyclic voltammogram under reaction conditions upon addition of TFA. Analysis of CV data from various combinations of the reaction components indicates that this new Fe species (**II**) likely contains both **L1** and trifluoroacetate ligands. Corresponding ions have also been observed by in situ HRMS.
- The reversible redox couple at -0.1 V (vs. Fc⁺/Fc), involving species **II** and **V**, has shown to be photoactive in CV experiments upon irradiation.
- Catalytic trifluoromethylation was detected even at high ligand **L1** loadings (30-50 mol%), where the concentration of non-ligated Fe species is expected to be minimal. This suggests that ligated Fe(III) trifluoroacetate species play also a role in promoting photodecarboxylation.

All data summarized and discussed in this introduction are provided in the revised Supplementary Information.

We believe that this comprehensive mechanistic hypothesis provides a more accurate representation of the processes occurring under reaction conditions and accounts for all the experimental data we have obtained.

Reviewer #1:

The manuscript entitled “Dual-Role Iron Species in Photoelectrocatalytic Radical Trifluoromethylation with Trifluoroacetates” reports a photoelectrocatalytic protocol for C(sp²)-H trifluoromethylation using readily available trifluoroacetates as CF₃ sources. The authors propose that in situ-generated Fe species play a “dual role” as both photo- and electrocatalysts, enabling decarboxylation and catalyst turnover without stoichiometric oxidants. In my view, this work represents an incremental rather than a conceptual advance over existing Fe-LMCT and electrophotocatalytic trifluoromethylation chemistry, and therefore does not meet the novelty and impact criteria for Nature Communications.

1. The combination of Fe-based LMCT photochemistry with electrochemical oxidation has already been demonstrated in several recent studies (e.g., Campbell et al., *Science* 2024, 383, 279; Chen et al., *Science* 2024, 384, 670; Qi et al., *J. Am. Chem. Soc.* 2023, 145, 24965). The current work does not introduce a clearly distinct mechanistic paradigm or reactivity principle beyond these precedents.

Unfortunately, I am afraid our perspective regarding the mechanistic paradigm of our transformation compared to the chemistry reported in these works differs from the concerns highlighted by this reviewer. The transformations developed by the groups of Nocera, Xuan & Mo and Wu represent very elegant and outstanding studies on the photoelectrochemical trifluoromethylations of organic substrates using trifluoroacetates or trifluoroacetic acid, as we have highlighted in the main text. The group of Nocera used silver salts to generate perfluoroalkyl radicals through the direct oxidation of the silver species at the electrode. The group of Xuan & Mo nicely engineered a WO₃ photoanode, enabling the selective oxidation of trifluoroacetates through an ion shielding electrode. The group of Jie Wu pioneered the direct oxidation of TFA at the anode to generate the trifluoromethyl radical. However, neither of these studies addresses Fe-based LMCT photochemistry. We would argue that our reactivity concept, which involves dual-role Fe active species, provides a complementary approach to these works.

2. The authors' previous report (*Angew. Chem. Int. Ed.* 2024, e202311984) already established Fe-LMCT photodecarboxylation of trifluoroacetates. The present study essentially replaces the chemical oxidant (K₂S₂O₈) with anodic oxidation to complete the catalytic cycle. While this modification improves atom economy, it constitutes an expected optimization rather than a conceptual breakthrough.

While the scientific community recognizes that enabling technologies can lead to more sustainable transformations, replacing a chemical oxidant with anodic oxidation is far from trivial. We hope that the more extensive mechanistic study presented in the revised manuscript clarifies this point. As highlighted in the introduction of this letter, the role of TFA is far from innocuous, because it significantly alters the catalyst's speciation under reaction conditions, facilitating a successful transformation. The necessity of TFA to electrochemically stabilize species **II** and **V** further distinguishes the CF₃-radical generation mechanism from our prior work, in which trifluoroacetate photodecarboxylation was carried out under basic conditions, suggesting that a different reactivity pathway may have been operative in that case.

Moreover, Motornov et al. (Angew. Chem. Int. Ed. 2025, 64, e202504143) have employed a similar iron catalyst in photoelectrochemical catalysis with a broader substrate scope.

I believe our perspective on this matter also differs from the reviewer's statement. Based on our new experimental data, which include CV, UV-Vis spectroscopy and in situ HRMS studies, we propose that our transformation operates through an intricate photoelectrocatalytic manifold. In our system, the ligand plays a critical role in enabling the catalyst to achieve the proper speciation, providing photoactive species that facilitate decarboxylation, and redox catalysts that mediate oxidative turnover of non-ligated iron intermediates. In the absence of the ligand, no Fe species exhibiting reversible CV are detected, and no reaction is observed in the absence of a redox mediator. While the reaction described by Motornov *et al.* proves effective for various fluoroalkylation reactions (including a single example of trifluoromethylation), it contrasts sharply with our findings. Their study demonstrates Fe carboxylate intermediates with reversible CV in the absence of ancillary ligands, underscoring the unique speciation of Fe trifluoroacetates in the presence of TFA. Our reactivity concept has enabled the trifluoromethylation of electron-rich, easily oxidizable substrates such as pyrroles, indoles, and electron-rich simple arenes. In contrast, when applying the method developed by Motornov *et al.* to these substrates, we were unable to obtain the corresponding trifluoromethylated products (see SI, page 56). This highlights the broader scope of our approach and its orthogonality to existing methodologies.

3. The mechanistic analysis is limited to cyclic voltammetry and UV–vis spectroscopy. No direct or quantitative evidence supports the coexistence of distinct Fe species responsible for photo- and electrocatalytic functions. Operando or time-resolved studies would be necessary to substantiate the proposed dual-catalyst mechanism.

In the revised manuscript and Supplementary Information, we have conducted a series of extensive and complementary experiments to support our more comprehensive mechanistic proposal. This updated study now includes a new set of control experiments, cyclic voltammetry measurements under varying conditions (including irradiation), as well as additional UV-Vis spectroscopic data and in situ HRMS analyses. Together, we believe these provide stronger and more compelling evidence for the validity of our proposed mechanistic hypothesis.

4. The substrate scope largely overlaps with known photoredox and electrophotocatalytic CF₃-introduction protocols. The functionalization of a few drug-like molecules (e.g., caffeine, melatonin) is insufficient to raise the overall impact to the level expected for Nature Communications.

The trifluoromethylation reaction is of central significance in numerous applied contexts and has therefore been the subject of extensive investigation. Although radical trifluoromethylation using classical reagents (e.g., Togni and Langlois) is well established, we argue that the deployment of trifluoroacetates or trifluoroacetic acid as trifluoromethylating agents for the derivatization of C(sp²)–H bonds remains comparatively underexplored. To our knowledge, only a limited number of studies, most notably those reported by the groups of Nocera, Wu, and Mo, which we have discussed in the text, have demonstrated such reactivity. In most cases, the scope of our transformation is broader than that of these methodologies. Our protocol is compatible not only with electron-deficient and electron-neutral substrates but also with more challenging electron-rich (hetero)arenes, which are typically susceptible to oxidative degradation. A quick survey of our reaction scope reveals at least 12 structurally diverse scaffolds bearing distinct substitution patterns, including the synthesis of trifluridine (one of the top-selling small-molecule pharmaceuticals globally) and the successful functionalization of at least four drug-like substrates.

Reviewer #2 (Remarks to the Author):

In this work, Francisco Juliá-Hernández et al. report a very elegant strategy for the photoelectrocatalytic radical trifluoromethylation of arenes using easily accessible alkali trifluoroacetate salts and TFA. This approach is highly powerful and could have a significant impact on the pharmaceutical and agrochemical industries by providing a more sustainable route to fluorinated molecules. This study builds upon the authors' 2024 *Angewandte Chemie* paper, in which they used a stoichiometric oxidant for a Minisci-type trifluoromethylation reaction. Clearly, the present work represents a greener approach, as it solves a long-standing problem in nucleophile cross-coupling: the use of stoichiometric amounts of reactive and aggressive oxidants, which often limit functional group tolerance and the overall reaction scope.

We appreciate the reviewer's comments recognizing the novelty and impact of our work.

The authors provide an elegant solution to this issue by employing dual iron electrocatalysis and photocatalysis via a LMCT (ligand-to-metal charge transfer). They found that a "ligand-free" iron trifluoroacetate species acts as a photocatalyst, generating the trifluoromethyl radical via light-induced charge-transfer homolysis and decarboxylation. Meanwhile, tris(bipyridine)-coordinated iron serves as an electron mediator in the electrocatalytic step. The authors present thorough mechanistic studies supported by UV-Vis spectroscopy, cyclic voltammetry (CV), and chronoamperometry, as well as hydrogen detection, all of which are discussed in detail in the manuscript and the Supporting Information (SI). The SI is of good quality and complete.

We thank the reviewer for acknowledging our efforts to elucidate the mechanism underlying this transformation. As noted in the introduction of this letter, we have expanded our mechanistic studies by extending our electrochemical and spectroscopic analyses to include conditions in the presence of trifluoroacetic acid (TFA). These studies revealed iron species bearing ligand **L1** and trifluoroacetate groups that are electrochemically stabilized by TFA. Notably, these species may serve a dual function in the reaction by promoting photodecarboxylation and facilitating redox turnover of non-ligated iron trifluoroacetate intermediates. Consequently, we propose a revised mechanistic framework involving two plausible productive pathways, each consistent with our experimental observations. All new data is summarized in the revised Supplementary Information and discussed in the revised manuscript.

I strongly recommend the publication of this article, as it will be an excellent contribution for the readers of Nature Communications.

Minor comments and suggestions:

-In Figure E6c, please correct the spelling of “voltajes” (should be “voltages”).

This has been amended.

-I would also like the authors to comment and discuss how the pyridyl radical, generated after the addition of the trifluoromethyl radical, undergoes oxidation, rearomatization, and proton release. Does this process occur directly at the electrode surface, or is the iron(III)L3 complex again the oxidant in this step?

We thank the reviewer for their valuable comments and suggestions. Indeed, following the addition of the CF₃ radical, the resulting cyclohexadienyl-type radical undergoes oxidation, proton release, and rearomatization. The efficiency of the oxidation step may vary depending on the nature of the substrate. In the revised manuscript, we have included the possibility that [Fe(L1)₃]³⁺ species could mediate the oxidative rearomatization in the bulk of the solution, in addition to direct oxidation at the anode.

-Additionally, I suggest that the authors discuss the reaction's efficiency at different ratios of the iron; L is used, as this will naturally affect the balance between “ligand-free” iron and tris(bipyridine) iron species.

We appreciate the reviewer’s suggestion. In line with our updated mechanistic proposal, decarboxylation can occur both from ligated Fe trifluoroacetate species (**V**) and from non-ligated intermediates (**IV**), the latter of which was initially proposed in the first version of the manuscript. This conclusion stems from an extensive investigation of Fe speciation in the presence of TFA, which revealed the formation of ligated Fe trifluoroacetate species that were found to be photoactive. Accordingly, when performing the photoelectrocatalytic trifluoromethylation under optimized conditions, but with increased loading of **L1** (30 and 50 mol%), we observed favorable reactivity, which likely results from the preferential formation of ligated species, such as **II**, which favored pathway A in our revised mechanistic proposal. Interestingly, when using 50 mol% of **L1**, the yield of the product was lower, likely due to the increased formation of Fe trisbipyridine-type

species (III), which are not directly involved in the decarboxylation of trifluoroacetates. These experiments have been included in the revised Supplementary Information.

entry	L1	CF ₃ -yield (bis)
1	10 mol %	76%
2	30 mol %	90%(9)
3	50 mol%	36%

-Finally, what is the effect of the bipyridine ligand structure (e.g., tert-butyl, Me or unsubstituted bpy) on the electron mediator properties?

As outlined in the introduction of this letter, the mechanistic pathway for our photoelectrocatalytic trifluoromethylation reaction is more complex than initially anticipated, involving ligated Fe trifluoroacetate species that act as bifunctional catalysts. These species both serve as redox mediators, facilitating the turnover of non-ligated Fe trifluoroacetate intermediates, and promote photodecarboxylation. In response to the reviewer's suggestion, we tested several electronically distinct bipyridine-type ligands and studied Fe speciation via cyclic voltammetry. However, none of the tested ligands resulted in more reactive conditions. Given the complexity of the reaction pathway, it is challenging at this stage to fully rationalize the observed trends solely based on the electronic properties of the ligands. This is due to the fact that each ligand class likely leads to distinct speciation, with intermediates that exhibit varying reactivity toward photodecarboxylation and different redox potentials. We have included the relevant data in the revised Supplementary Information.

entry	deviation	yield of 2
1	L1	76%
2	L2	63%
3	L3	52%
4	L4	0%
5	L5	46%
6	L6	29%

L1

L2

L3

L4

L5

L6

CV study with **L1**:

Cyclic voltammograms of: **(i)** Fe(OTf)₂ (10 mM) [grey trace]; **(ii)** Fe(OTf)₂ (10 mM) and NaO₂CCF₃ (600 mM) [green trace], **(iii)** Fe(OTf)₂ (10 mM) and **L1** (10 mM) [blue trace], **(iv)** Fe(OTf)₂ (10 mM), **L1** (10 mM) and NaO₂CCF₃ (600 mM) [orange trace]. Every independent measurement was done in the presence of trifluoroacetic acid (TFA, 130 mM).

CV study with L2:

Cyclic voltammograms of: **(i)** Fe(OTf)₂ (10 mM), L2 (10 mM) and TFA (130 mM) [blue trace]; **(ii)** Fe(OTf)₂ (10 mM), L2 (10 mM), NaO₂CCF₃ (600 mM) and TFA (130 mM) [purple trace].

CV study with L3:

Cyclic voltammograms of: **(i)** Fe(OTf)₂ (10 mM) and L3 (10 mM) [black trace]; **(ii)** Fe(OTf)₂ (10 mM), L3 (10 mM) and TFA (130 mM) [blue trace]; **(iii)** Fe(OTf)₂ (10 mM), L2 (10 mM), NaO₂CCF₃ (600 mM) and TFA (130 mM) [purple trace].

CV study with L4:

Cyclic voltammograms of: **(i)** Fe(OTf)₂ (10 mM) and L4 (10 mM) [black trace]; **(ii)** Fe(OTf)₂ (10 mM), L4 (10 mM) and TFA (130 mM) [blue trace]; **(iii)** Fe(OTf)₂ (10 mM), L4 (10 mM), NaO₂CCF₃ (600 mM) and TFA (130 mM) [purple trace].

CV study with L5:

Cyclic voltammograms of: **(i)** Fe(OTf)₂ (10 mM) and L5 (10 mM) [black trace]; **(ii)** Fe(OTf)₂ (10 mM), L5 (10 mM) and TFA (130 mM) [blue trace]; **(iii)** Fe(OTf)₂ (10 mM), L5 (10 mM), NaO₂CCF₃ (600 mM) and TFA (130 mM) [purple trace].

CV study with L6:

Cyclic voltammograms of: (i) Fe(OTf)₂ (10 mM) and L6 (10 mM) [black trace]; (ii) Fe(OTf)₂ (10 mM), L6 (10 mM) and TFA (130 mM) [blue trace]; (iii) Fe(OTf)₂ (10 mM), L6 (10 mM), NaO₂CCF₃ (600 mM) and TFA (130 mM) [purple trace].

Reviewer #3 (Remarks to the Author):

In this communication, Fernández-García et. al. report a photoelectrocatalytic methodology for the decarboxylation of cheap and abundant TFA using Fe catalysis, producing trifluoromethyl radicals that may be harnessed for the trifluoromethylation of (hetero)arenes. This work is a continuation of their 2024 ACIE manuscript (Angew. Chem. Int. Ed. 2024, 63, e202311984; Ref. 11 in the current work) in which the main advance is moving away from a chemical oxidant (K₂S₂O₈) toward electrochemical oxidation. Thus, this is the most distinguishing point of the paper. It certainly is not a big leap to do so but there are a few notable points. While photoelectrocatalysis has previously been demonstrated in Fe-mediated fluoroalkylation from fluoroalkyl carboxylates (Ackermann, Angew. Chem. Int. Ed. 2025, 64, e202504143, ref. 40 in the current work), the current work is distinguished by perhaps broader applicability towards electron-rich functionalities, as well as a potentially unique reaction mechanism. The demonstrated reaction scope appears nearly identical to the author's prior work (ref. 11), although the tolerance of electron-rich functionality (substrate 2) is notable under electrochemical conditions where anodic overoxidation is a possibility. In light of the above points, I recommend publication after addressing the following comments, which mainly concern 1) clarification of the reaction mechanism and 2) differentiation of the methodology from prior work.

We would like to take this opportunity to thank this reviewer for his/her valuable feedback, which has been instrumental in enhancing the impact of our work. The elucidation of a more comprehensive mechanistic hypothesis now provides a more accurate representation of the processes occurring under the reaction conditions. This has been summarized at the beginning of this letter, and it has also been incorporated into the revised manuscript and Supplementary Information.

1. The redox features in the cyclic voltammograms are reported vs a Ag/AgCl reference. The authors are encouraged to report the potentials vs ferrocene or SCE instead, as these values are more utilized and easily comparable in non-aqueous systems.

Ferrocene has been used to report cyclic voltammograms in both the revised manuscript and Supplementary Information.

2. The authors state: “whereas the redox signals corresponding to complex II remained stable, displaying a higher anodic than cathodic current, suggestive of electrocatalytic behavior.” In reference to Figure S20. Upon inspection of Figure S20, it appears the CV was slightly misinterpreted. Peak anodic and peak cathodic currents should not be read from absolute value on the y-axis. Please do some reading in the most rudimentary electrochemical textbooks to properly interpret the CVs. When reading from the absolute values, the peak anodic current of the last scan (dark orange line) is 25 μA and the peak cathodic current is $-11 \mu\text{A}$, which is perhaps why the authors made the above statement. However, the baseline in the forward wave is about 5 mA while that of the reverse wave is ca. 9 mA, meaning that the peak anodic current is 20 μA and cathodic current $-20 \mu\text{A}$ when read from these values, indicating a fully reversible wave, as expected for species II. Absolutely no evidence for electrocatalysis can be gleaned from this CV behavior. These are the types of sloppy mistakes this field is prone to ... electrochemistry is a field unto itself. It needs to be done properly even if this is an “organic” manuscript.

We sincerely regret this misinterpretation. This explanation is no longer in the manuscript as the mechanistic investigation has been further improved.

3. The assignment of species III in the CVs is ambiguous. It’s irreversibility and similarity to a CV of $\text{Fe}(\text{OTf})_2 + \text{NaO}_2\text{CCF}_3$ suggest it is a TFA ligated Fe species, but a few questions remain. Figure 2B should include concentrations of analytes in the figure caption or figure legend. If the orange trace and green trace are equimolar Fe concentrations, why does the orange trace feature a boosted III redox wave?

We thank the reviewer for these comments. In Fig. 2B of the original submission, the total iron concentration for both orange and green curves was not the same. Indeed, at equimolar concentrations, this boost is no longer observed:

Cyclic voltammograms of: **(i)** $\text{Fe}(\text{OTf})_2$ (10 mM) [grey trace]; **(ii)** $\text{Fe}(\text{OTf})_2$ (10 mM) and NaO_2CCF_3 (600 mM) [green trace], **(iii)** $\text{Fe}(\text{OTf})_2$ (10 mM) and **L1** (10 mM) [blue trace], **(iv)** $\text{Fe}(\text{OTf})_2$ (10 mM), **L1** (10 mM) and NaO_2CCF_3 (600 mM) [orange trace]. Every independent measurement was done in the presence of trifluoroacetic acid (TFA, 130 mM).

Figure S20 shows that this wave diminishes significantly upon cycling, suggesting that the chemical step following oxidation is depleting the precursor to III overtime. Can the authors also cycle a CV of $\text{Fe}(\text{OTf})_2 + \text{NaO}_2\text{CCF}_3$ similarly to Figure S20? Does the redox wave at -0.35 V also diminish in intensity?

We performed this experiment and observed that the redox wave diminished in intensity in the absence of TFA, with the redox wave now occurring at 0.15 V vs Fc^+/Fc .

Sequential cyclic voltammograms of $\text{Fe}(\text{OTf})_2$ (10 mM) + NaO_2CCF_3 (600 mM).

If equimolar amounts of Fe and TFA are used, does the addition of L1 truly boost the current associated with III in the first scan, as Figure 2B suggests? This would mean that there is some Fe species in solution that contains both TFA and L1.

When equimolar amounts of $\text{Fe}(\text{OTf})_2$ and sodium trifluoroacetate are employed, the addition of **L1** results in the formation of a complex mixture, consisting of $\text{Fe}(\text{OTf})_2$, $[\text{Fe}(\text{L1})_3]^{2+}$ and solvated Fe trifluoroacetate species, as observed in the cyclic voltammogram.

*Cyclic voltammograms of: (i) $\text{Fe}(\text{OTf})_2$ (10 mM) and NaO_2CCF_3 (10 mM) [black trace]; (ii) $\text{Fe}(\text{OTf})_2$ (10 mM), NaO_2CCF_3 (10 mM) and **L1**(10 mM) [red trace].*

In any case, as noted in the introduction of this letter, we have expanded our mechanistic studies by extending our electrochemical and spectroscopic analyses to include conditions in the presence of trifluoroacetic acid (TFA). These studies revealed iron species bearing ligand **L1** and trifluoroacetate groups that are electrochemically stabilized by TFA. Notably, these species may serve a dual function in the reaction by promoting photodecarboxylation and facilitating redox turnover of non-ligated iron trifluoroacetate intermediates. Consequently, we propose a more comprehensive mechanistic framework involving two plausible productive pathways, each consistent with our experimental observations. This has been summarized at the beginning of this letter, and it has also been incorporated into the revised manuscript and Supplementary Information.

4. The above point is convoluted by the fact that the authors observe near quantitative formation of **II** via Beer-Lambert analysis, ruling out Fe species that contain both TFA and L1. If this is true, it is unclear why L1 would boost the anodic current of **III** on the first scan, as no **IV** is present in solution yet to mediate the oxidation of **III** to **V**.

We greatly appreciate these suggestions, which have prompted us to reevaluate the mechanistic investigation in the presence of trifluoroacetic acid. In the presence of TFA, the formation of the trisbipyridine-type iron species has been calculated to be less than 23%. Therefore, it is highly plausible that Fe species containing both **L1** and trifluoroacetate ligands are formed in significant concentrations, as suggested by the reviewer. Indeed, the corresponding ions have also been observed by in situ HRMS. The new experimental data have been added to the revised Supplementary Information and discussed in the main text.

The claim that quantitative **II** is formed suggests that adding too much L1 should completely shut off reactivity, as no Fe would be left to ligate to TFA. It does not appear that the authors included an optimization of L1 concentration, unless I am mistaken. If 30 mol% L1 or greater is employed, all the Fe should be sequestered as **II**, and the yield should drop. Can the authors demonstrate this? Recent work published on the rxiv (<https://doi.org/10.26434/chemrxiv-2025-djp0r>) showcases TFA decarboxylation from well-defined (dien)FeTFA complexes. It is possible that the photoactive species can still contain a mixture of TFA and bpy ligands. This, of course, would be supported by the presence of product even when employing, say, 50 mol% ligand.

Once again, we would like to thank this reviewer for the comments, which have been key in directing our revised mechanistic investigation. In line with our more comprehensive mechanistic proposal, decarboxylation can occur both from ligated Fe trifluoroacetate species (**V**) and from non-ligated intermediates (**IV**), the latter of which was initially proposed in the first version of the manuscript. This conclusion stems from an extensive investigation of Fe speciation in the presence of TFA, which revealed the formation of ligated Fe trifluoroacetate species that were found to be photoactive. Accordingly, when performing the photoelectrocatalytic trifluoromethylation under optimized conditions, but with increased loading of **L1** (30 and 50 mol%), we observed favorable reactivity, which likely results from the preferential formation of ligated species, such as **II** and **III**,

which favored pathway A in our revised mechanistic proposal. Interestingly, when using 50 mol% of **L1**, the yield of the product was lower, likely due to the increased formation of Fe trisbipyridine-type species, which are not directly involved in the decarboxylation of trifluoroacetates. These experiments have been included in the revised Supplementary Information.

entry	L1	CF₃ -yield (bis)
1	10 mol %	76%
2	30 mol %	90%(9)
3	50 mol%	36%

5. Notably, the authors demonstrate that Ackermann's conditions (ref 40) result in 0 percent yield on substrate 2, showcasing the current methodology's tolerance towards more electron-rich substrates. However, the authors attribute this to the role **L1** plays in the proposed mechanism. Is it possible that **L1** simply serves to decrease the required oxidation potential to access Fe(III)? Thus, the yield is 0 under Ackermann's conditions due to direct oxidation of 2 at the anode. Ackermann demonstrates successful trifluoromethylation of caffeine without a ligand, which is harder to oxidize. Under the current conditions, if **L1** is excluded, can caffeine still be engaged? If not, this is likely due to the low cell potential (2.06 V), if this is raised, or constant current electrolysis is utilized, I assume caffeine can be engaged without **L1**. Please perform this experiment.

Experiments using caffeine as the substrate were performed in the absence of **L1**, under constant cell potential, and at a constant current of 4 mA, resulting in very low yields of the corresponding trifluoromethylated caffeine (15% and 2%, respectively). These results indicate that caffeine cannot be effectively engaged under these conditions, suggesting that our reaction pathway differs significantly from the one operating under Ackermann's conditions. The data have been included in the revised Supplementary Information.

entry	conditions	CF ₃ -yield
1	$E_{\text{cell}} = 2.06 \text{ V}$	15%
2	$i = 4.0 \text{ mA}$	2%

6. The use of mediators instead of L1 restores reactivity, including non Fe-containing mediators such as magic blue; this is a notable result. However, species III is still easier to oxidize at an electrode than these mediators, so it is unclear as to why the harder to oxidize species would serve as an electron shuttle. A separate hypothesis is that the redox mediator may help to oxidize arene radical intermediates in solution to the cation, which facilitates rearomatization after radical addition following proton loss.

We completely agree with the reviewer in this comment. Indeed, in our revised mechanism, we propose that species **V**, containing both **L1** and trifluoroacetate ligands, with an $E_{1/2}$ at -0.1 V (vs. Fc^+/Fc), may play a role in mediating the oxidation of non-ligated Fe trifluoroacetate species (**I**) with $E_{p/2}$ at $+0.1 \text{ V}$ (vs. Fc^+/Fc). Additionally, we found that, in the absence of **L1**, ferrocene can mediate this oxidation, as its cyclic voltammogram closely resembles that of species **V**. In the case of Fe trisbipyridine-type complexes or acetylferrocene, which exhibit reversible redox couples with oxidation potentials greater than those of non-ligated Fe trifluoroacetate intermediates, but within the range of the applied potential, their role may be related to facilitating the rearomatization of the resulting cyclohexadienyl-type radicals after CF_3 addition in solution. The efficiency of the oxidation step may vary depending on the substrate. In the revised manuscript, we have included the possibility that $[\text{Fe}(\mathbf{L1})_3]^{3+}$ species could mediate the oxidative rearomatization in the bulk of the solution, in addition to direct oxidation at the anode. The role of magic blue, which is harder to oxidize compared to the other redox mediators used, under reaction conditions without **L1**, remains difficult to clarify. While magic blue could be involved in photoinduced events, its precise role in the transformation remains speculative.

7. Altogether, the mechanistic hypothesis proposed in Figure 2D is perhaps plausible, but many uncertainties remain. To me, it is clear that L1 ligates Fe(II), making it easier to oxidize to Fe(III) which is required for LMCT. The fact that no yield is observed in the absence of L1, even though no ligands are required in other work (ref. 40) is because, in this work, the potential at the anode is low. This helps facilitate reactions with electron-rich substrates. Presumably, the use of other bipyridine derivatives, or phenanthroline derivatives, can further modulate the potential required for Fe(II) oxidation. How are the CVs changed when more or similar electron rich bpy derivatives are used (R = NMe₂, tBu, Me, etc) or when more electron-poor derivatives are used (R = CF₃, CO₂Me, Cl, F, etc.)? Can reactivity be improved with a different ligand? If the required oxidation potential is moved too anodic with more electron poor bpy derivatives, does this completely shut off reactivity in the utilized potential window? Mechanistic experiments with varied bpy ligands should be carried out to provide more insight into the proposed mechanism.

As outlined in the introduction of this letter, the mechanistic pathway for our photoelectrocatalytic trifluoromethylation reaction is more complex than initially anticipated, involving ligated Fe trifluoroacetate species that act as bifunctional catalysts. These species both serve as redox mediators, facilitating the turnover of non-ligated Fe trifluoroacetate intermediates, and promote photodecarboxylation. In response to the reviewer's suggestion, we tested several electronically distinct bipyridine-type ligands and studied Fe speciation via cyclic voltammetry. However, none of the tested ligands resulted in more reactive conditions. Given the complexity of the reaction pathway, it is challenging at this stage to fully rationalize the observed trends solely based on the electronic properties of the ligands. This is due to the fact that each ligand class likely leads to distinct speciation, with intermediates that exhibit varying reactivity toward photodecarboxylation and different redox potentials. We have included the relevant data in the revised Supplementary Information.

entry	deviation	yield of 2
1	L1	76%
2	L2	63%
3	L3	52%
4	L4	0%
5	L5	46%
6	L6	29%

L1

L2

L3

L4

L5

L6

CV study with **L1**:

Cyclic voltammograms of: **(i)** Fe(OTf)₂ (10 mM) [grey trace]; **(ii)** Fe(OTf)₂ (10 mM) and NaO₂CCF₃ (600 mM) [green trace], **(iii)** Fe(OTf)₂ (10 mM) and **L1** (10 mM) [blue trace], **(iv)** Fe(OTf)₂ (10 mM), **L1** (10 mM) and NaO₂CCF₃ (600 mM) [orange trace]. Every independent measurement was done in the presence of trifluoroacetic acid (TFA, 130 mM).

CV study with L2:

Cyclic voltammograms of: **(i)** Fe(OTf)₂ (10 mM), L2 (10 mM) and TFA (130 mM) [blue trace]; **(ii)** Fe(OTf)₂ (10 mM), L2 (10 mM), NaO₂CCF₃ (600 mM) and TFA (130 mM) [purple trace].

CV study with L3:

Cyclic voltammograms of: **(i)** Fe(OTf)₂ (10 mM) and L3 (10 mM) [black trace]; **(ii)** Fe(OTf)₂ (10 mM), L3 (10 mM) and TFA (130 mM) [blue trace]; **(iii)** Fe(OTf)₂ (10 mM), L2 (10 mM), NaO₂CCF₃ (600 mM) and TFA (130 mM) [purple trace].

CV study with L4:

Cyclic voltammograms of: **(i)** Fe(OTf)₂ (10 mM) and L4 (10 mM) [black trace]; **(ii)** Fe(OTf)₂ (10 mM), L4 (10 mM) and TFA (130 mM) [blue trace]; **(iii)** Fe(OTf)₂ (10 mM), L4 (10 mM), NaO₂CCF₃ (600 mM) and TFA (130 mM) [purple trace].

CV study with L5:

Cyclic voltammograms of: **(i)** Fe(OTf)₂ (10 mM) and L5 (10 mM) [black trace]; **(ii)** Fe(OTf)₂ (10 mM), L5 (10 mM) and TFA (130 mM) [blue trace]; **(iii)** Fe(OTf)₂ (10 mM), L5 (10 mM), NaO₂CCF₃ (600 mM) and TFA (130 mM) [purple trace].

CV study with L6:

Cyclic voltammograms of: **(i)** Fe(OTf)₂ (10 mM) and L6 (10 mM) [black trace]; **(ii)** Fe(OTf)₂ (10 mM), L6 (10 mM) and TFA (130 mM) [blue trace]; **(iii)** Fe(OTf)₂ (10 mM), L6 (10 mM), NaO₂CCF₃ (600 mM) and TFA (130 mM) [purple trace].

8. The reaction times should be included in Figure 3.

These have been included.

9. Substrates 10 and 11 do not contain sterically accessible coordinating nitrogen atoms, so the claim “Notably, the catalyst remained active in the presence of substrates containing potentially coordinating nitrogen atoms” is not supported.

This sentence has been removed from the revised manuscript.

10. The demonstrated scope is remarkably similar to the author’s previous work (ref. 11). Compare substrate 11 (this work) vs 17 (ref. 11), 8 (this work) vs 19 (ref. 11), 21 (this work) vs 33 (ref. 11), or 19 (this work) vs 29 (ref. 11). Even substrates highlighted as challenging due to their electron-rich nature (18) performed equally well, if not slightly better, in ref. 11 (substrate 24). Is there any substrate or substrate class that can be engaged in the current methodology that is not amendable to the previous conditions employed in ref. 11? Nonetheless, it is understood that elimination of a stoichiometric oxidant increases the atom economy of the transformation.

In response to the reviewer’s suggestion, we have conducted additional experiments to further expand the scope of the reaction. However, at this stage, we have not been able to obtain the corresponding products in synthetically useful yields (>50%). Some low-yielding substrates are highlighted on page 30 of the Supplementary Information. Nonetheless, we would like to emphasize that our trifluoromethylation reaction has been successfully applied to more than 12 different (hetero)cyclic scaffolds with various substitution patterns. Furthermore, we have successfully scaled up the reactions, obtaining trifluoromethylated caffeine (**19**) and trimethoxybenzene (**2**) in synthetically useful yields, an outcome that was not achievable under the conditions of our previous work.

11. The authors should check their assignment of the regioisomer depicted for substrate **12**, it is more likely that the radical addition occurs next to the carbonyl moiety.

We have isolated and fully characterized product **12**. We have amended the assignment of the regioisomer by direct comparison with the literature data of the isolated compound. We have added the characterization data to the revised Supplementary Information:

4,6-dimethyl-5-(trifluoromethyl)-2H-pyran-2-one (12)

12

Prepared following the general procedure C and starting from *4,6-dimethyl-2H-pyran-2-one* (62.1 mg, 0.5 mmol). The crude mixture was purified by flash column chromatography on a silica gel column, using a mixture of Cyclohexane/Et₂O (6:4) to provide **12** as a white solid (49 mg, 51% yield).

R_f = 0.42 (Cyclohexane/ Et₂O (6:4) (v/v)).

¹H NMR (600 MHz, CDCl₃): δ 5.92 (s, 1H), 2.33 (q, *J* = 2.9Hz, 3H), 6H), 2.26 (s, 3H) ppm.

¹³C NMR (151 MHz, CDCl₃) δ 164.1, 158.3, 158.2, 123.3 (q, *J* = 274.0 Hz), 111.2 (q, *J* = 31.3Hz), 108.4, 20.8 (q, *J* = 4.0 Hz), 19.9 ppm.

¹⁹F NMR (376 MHz, CDCl₃): δ -58.06 ppm.

HRMS (ESI, *m/z*) calcd. For C₁₂H₁₃F₃O₄ [M⁺]: 278.0766; found 278.0763.

Spectroscopic data is in agreement with previously reported literature data. (J. Cornella *et al.*, *J. Am. Chem. Soc.* **2023**, *145*, 25538)

12. If the redox wave for III is overlaid in Figure 6, it looks as if the yield begins to rise when accessing that couple, as opposed to the oxidation of the mediator. In fact, the yield plateaus even before reaching the E_{1/2} of Fe(II)L13 to Fe(III)L13. The yield at 0.5 V is also very comparable to the yield at 0.8 V, even though the current displayed under these two conditions are very different. This speaks to some of the remaining mechanistic ambiguity highlighted above. Please address.

In light of the new mechanistic studies conducted in the presence of TFA, the yield plateau is reached at an anodic potential slightly above the $E_{1/2}$ of the Fe-trisbipy pair for caffeine as the substrate. This suggests that, as mentioned above, the **III/VI** redox pair may play a role in the rearomatization step or in the turnover of the non-ligated Fe trifluoroacetate species in the bulk of the solution.

Graphical overlap between: (i) the cyclic voltammetry of $\text{Fe}(\text{OTf})_2$ (10 mM), **L1** (10 mM) and NaO_2CCF_3 (600 mM) (reaction mixture; left Y-axis); and (ii) the graphical representation of the relationship between anode potential (E_{anode}) vs yield of product **19** (right Y-axis).

Overall, we are grateful for the referee's comments which have been very useful in further improving the impact and presentation of our work.

“Dual-Role Iron Species in Photoelectrocatalytic Radical Trifluoromethylation with Trifluoroacetates”

(manuscript number NCOMMS-25-80469A)

Point-by-point Response to the Reviewers' Comments

Reviewer #1:

The authors have done careful revision and detailed response. The additional mechanistic and electrochemical studies substantially improve the clarity and rigor of the manuscript, and the revised interpretation of iron speciation and CV data is appropriate. Although my view that this work represents an incremental rather than conceptual advance relative to prior Fe-LMCT and photoelectrochemical trifluoromethylation chemistry largely remains, I acknowledge that the revised manuscript more clearly differentiates this system through ligand-controlled iron speciation, the critical role of TFA, and its compatibility with electron-rich substrates. I consider my major concerns addressed and recommend acceptance.

We sincerely appreciate the reviewer's comments, which support our revised mechanistic proposal and recognize our efforts to improve our work.

Reviewer #2:

In this resubmission, Francisco Juliá-Hernández and the co-authors have thoroughly addressed all of my questions and carefully considered my suggestions. The scientific quality of the work is very high, and the results are both timely and impactful.

We are grateful for the reviewer's evaluation, which has greatly helped us further enhance the impact and quality of our work.

I have the following optional comments that can improve the clarity of the work:

1. I found the manuscript structure somewhat challenging, particularly with respect to the separation of the mechanistic discussion across the scope section. I would suggest presenting the mechanistic considerations together, either before or after the scope, to improve clarity and readability.

We thank the reviewer for this suggestion. We agree that the structure of the manuscript is somewhat unconventional; however, we believe it achieves a balance between hypothesis, mechanistic investigation and synthetic analysis. We think that positioning key mechanistic experiments at the beginning of the manuscript is essential to support the reaction hypothesis. At the same time, moving the remaining mechanistic experiments, currently placed at the end, earlier in the text could make the mechanistic section too dense, before the synthetic potential of the transformation is properly addressed (another central aspect of our work). We therefore think that maintaining the current structure, introducing the scope after the hypothesis while leaving the additional mechanistic experiments at the end, offers the best compromise. In our opinion, this arrangement allows readers to engage with the work, whether interested in the mechanistic aspects or the synthetic potential.

2. At present, the mechanistic picture involves the interplay of multiple iron species that can undergo LMCT to generate photoactive iron trifluoroacetate species capable of delivering the trifluoromethyl radical. In this context, UV-vis titration experiments could be used to quantify the relative amounts of iron chelates bearing three, two, or one ligand. Did the authors explore this possibility?

We appreciate the reviewer's insightful and constructive suggestion. Indeed, we have identified several ligated and non-ligated Fe trifluoroacetate species able to promote the formation of CF_3 radicals by electrochemical measurements (CV) and in-situ HRMS. According to our experimental data, these species arise from an equilibrium formed under reaction conditions, as discussed in the manuscript. In response to the reviewer's suggestion, the quantification of the possible mono-, bis- and tris-ligated Fe species by UV-Vis titration experiments is not trivial. On one hand, the ligated Fe intermediates absorb in overlapping regions of the spectrum (see Fig S9 and S10 in the supplementary information), which hampers their unambiguous identification. On the other hand, we have been unable to isolate the corresponding discrete chelated Fe species because of their rapid equilibration in solution. Consequently, suitable pure standards are not available to determine the relative abundances of the individual intermediates.

3. Importantly, reaction yields are not necessarily correlated with the rates of elementary steps. I therefore suggest examining the stoichiometric reaction kinetics (in the absence of electricity and TFA) by trapping the CF_3 radical with TEMPO, which would facilitate interpretation of the relative efficiency of the various iron species involved in LMCT. For the kinetic analysis (e.g., by ^{19}F NMR) of the stoichiometric system, the authors could employ $\text{Fe}(\text{OTf})_3$ (1 equiv, 0.05M), $\text{CF}_3\text{CO}_2\text{Na}$ (>10 equiv), and systematically vary the ligand loading (0, 1, 2 and 3 equiv), collecting several time points at low conversion (<40%). Analysis of the resulting kinetic profiles could help identify which of the non-ligated, mono-ligated, or bis-bipyridine-ligated iron species is most efficient at generating the trifluoromethyl radical.

Studying stoichiometric reactions that involve elementary steps could provide useful information about the underlying catalytic process. In our catalytic system, we have shown that although a 1:1 ratio between Fe and **L1** is used, this does not lead to the formation of a discrete mono-ligated intermediate. Instead, an equilibrium of different iron species is formed, with TFA playing a key role in the speciation. Therefore, we think that although the study of stoichiometric reactions starting from Fe(III) and different ligand loadings would provide information about the decarboxylation process, this might not be very representative of our reaction system in the absence of TFA. Additionally, we do not expect CF_3 radical trapping by TEMPO to be highly efficient under these conditions, as shown in our previous work (*Angew. Chem. Int. Ed.* **2024**, *63*, e202311984). It is also important to consider that TEMPO can act as an effective oxidant, promoting the Fe(II) to Fe(III) turnover, which may lead to non-stoichiometric processes and thereby complicate the analysis. Such behavior has recently been demonstrated for ligated Fe species by Ala Bunescu and co-workers in an elegant decarboxylative oxygenation reaction of aliphatic carboxylic acids (*Angew. Chem. Int. Ed.* **2024**, *63*, e202403292).

Nevertheless, we believe that the reviewer's suggestion regarding the relative efficiencies of the different Fe species in solution in promoting photoinduced LMCT decarboxylation to generate CF_3 radicals is pertinent. We investigated the two photodecarboxylation pathways proposed in our mechanistic hypothesis by monitoring the photoreduction of in situ-formed ligated and non-ligated Fe(III) trifluoroacetate species, corresponding to intermediates **IV** and **V**, respectively. To this end, mixtures of $\text{Fe}(\text{OTf})_3$, NaO_2CCF_3 and TFA in acetonitrile were prepared in the presence and absence of **L1**, and their evolution under irradiation was followed by UV-Vis absorption spectroscopy.

Photodecarboxylation of non-ligated Fe(III) trifluoroacetate species (absence of L1)

(a) UV-Vis spectra of a mixture of Fe(OTf)₃ (10 mM), NaO₂CCF₃ (600 mM) and TFA (130 mM) in dry acetonitrile, at different irradiation times (irradiation at 405 nm) [*orange gradient*]; (b) UV-Vis spectrum of a mixture of Fe(OTf)₂ (10 mM), NaO₂CCF₃ (600 mM) and TFA (130 mM) [*purple line*].

After 30 minutes of irradiation, the non-ligated Fe(III) trifluoroacetate species associated with intermediate **IV** was completely consumed, as evidenced by the disappearance of the absorption band at 350 nm. Concomitantly, Fe(II) trifluoroacetate species were formed, as confirmed by comparison with the purple reference spectrum. The photoreduction of Fe(III) to Fe(II) arises from the decarboxylative LMCT process.

Photodecarboxylation of ligated Fe(III) trifluoroacetate species (with L1)

(a) UV-Vis spectra of a mixture of Fe(OTf)₃ (10 mM), L1 (10 mM), NaO₂CCF₃ (600 mM) and TFA (130 mM) in dry acetonitrile, at different irradiation times (irradiation at 405 nm) [*orange gradient*]; (b) UV-Vis spectrum of a mixture of Fe(OTf)₂ (10 mM), L1 (10 mM), NaO₂CCF₃ (600 mM) and TFA (130 mM) [*purple line*].

Photoreduction of the ligated Fe(III) trifluoroacetate species is evidenced by the emergence of an absorption band in the 500–600 nm region, which is attributed to Fe(II)/L1 intermediates based on comparison with the purple reference spectrum.

Photoreduction of Fe(III) species in the presence and in the absence of ligand **L1**. The inset shows zoomed data up to 0.5 h, indicating the photoreduction of both Fe(III) species after 30 min.

Notably, in the presence of **L1** the LMCT photodecarboxylation proceeds much more slowly than for the corresponding non-ligated species, requiring approximately 20 h to reach completion. Taken together, this study indicates that the photodecarboxylation of trifluoroacetates proceeds much more rapidly from non-ligated Fe(III) species, which are involved in path B of the proposed mechanism.

These new experiments and data have been incorporated into the revised supplementary information.

Overall, as stated previously, I believe this work is of excellent quality, and I strongly recommend it for publication in Nature Communications.

We want to take the opportunity to thank the reviewer once again for the feedback and suggestions provided.

Reviewer #3:

In response to reviewer comments, the authors of this work have drastically changed the mechanistic interpretation of the reaction mechanism, aligned with suggestions made in my original review of the manuscript. The effort the authors put into this mechanistic reevaluation is appreciated, and in my opinion, the mechanism is sounder than the original draft. However, certain questions remain. Chief among them is the role of Path B. Path B seems almost entirely irrelevant, and the reaction can be purely described by Path A, which is no different than many electrophotocatalytic schemes reported in the literature. I agree with Reviewer #1 that the current work is an incremental advance of the authors' prior work, with many of the defining principles of the work already having much precedent (electrochemical turnover of Fe perfluoroalkyl carboxylates demonstrated by Ackermann, substrate scope identical to authors' prior work), as I also stated in my original review. Thus, paramount to the novelty of this work is the proposed distinct mechanistic paradigm involving "dual-role" iron species. It appears to me that the authors are clinging to this mechanistic interpretation to preserve the novelty of the current work, but all signs point to a normal electrophotocatalytic pathway (i.e. Path A) that is not altogether a new concept.

We thank the reviewer for the suggestions and for recognizing our efforts to establish a more robust mechanistic proposal.

In the original draft, Path A was not discussed by the authors as a possibility. Rather, all reactivity was described to an electrocatalytic turnover of I to IV (III to V in the original draft) mediated by FeL₃. Many of my comments (example: comment #s 3, 4, 7, 12) drove after the point that reactivity could be enabled entirely by heteroleptic FeLnTFAM complexes that (i) undergo Fe(II) \leftrightarrow Fe(III) redox at an electrode and (ii) undergo Fe(III) \leftrightarrow Fe(II) redox via photochemical LMCT. With new experiments, the authors now include this mechanism as Pathway A in the current work. This represents a familiar LMCT electrophotocatalytic cycle, for example, identical to the scheme proposed by Nocera (Ag(I) \leftrightarrow Ag(II) redox at an electrode and Ag(II) \leftrightarrow Ag(I) redox via photochemical LMCT). Pathway B remains similar to the original hypothesis, but it is not sufficiently substantiated.

We appreciate the reviewer's valuable comments. In response, we have carried out new kinetic experiments that, in our view, clarify the role of path B in our reaction and

highlight the importance of ligand addition in enabling a broader reaction scope. We have also provided experimental evidence supporting the presence of non-ligated Fe trifluoroacetate species in solution, and their rapid photodecarboxylation under irradiation. We believe that these new data provide further support for the dual role of the ligated Fe trifluoroacetate species (**V**) in promoting photoinduced decarboxylation and facilitating the turnover of non-ligated Fe species acting as a redox mediator.

These new experiments have been incorporated into the second revised version of the manuscript and supplementary information and are discussed below in response to the specific questions raised by this reviewer.

(1) Pathway B invokes chemical oxidation of **I** by **V**. inspection of the CV data shows that **I** is harder to oxidize than **II**, meaning the **V/II** couple should not be capable of inducing oxidation of **I** to **IV**. In fact, oxidation of **I** is ~200 mV uphill from oxidation of **II**, and practically zero oxidative current for **IV/I** exists at the $E_{1/2}$ of **V/II**. Given this, why do the authors argue that the **IV/II** couple is relevant under catalytic conditions? Furthermore, HRMS data do not even support the existence of ligandless FeTFA salts.

We would like to thank the reviewer for a particular insightful statement in the previous revision, which we found to be very helpful: “*species **III** is still easier to oxidize at an electrode than these mediators, so it is unclear as to why the harder to oxidize species would serve as an electron shuttle.*”. This comment prompted us to carefully reevaluate our mechanistic proposal through multiple additional experiments. As it is described in the literature, redox mediators act as electron shuttles between the substrate and the electrode. In indirect anodic oxidation, the mediator usually has a lower redox potential than the substrate, enabling milder oxidative conditions with improved chemoselectivity. In our proposal, mediator **V** has a lower redox potential than “substrate” **I**. Therefore, electron transfer in solution occurs against a potential gradient, which is possible when a subsequent thermodynamically favorable step drives the equilibrium forward. In our system, there is a 200 mV uphill gap in the mediated oxidation of species **I** to **IV** as described by the reviewer, a process that is driven by the rapid, irreversible photoinduced LMCT decarboxylation at **IV** (as demonstrated below). There are numerous precedents in the literature reporting mediated redox reactions with gaps even higher than 200 mV.

Examples of these can be found in the selected publications listed below:

- Shannon Stahl and co-workers: *J. Am. Chem. Soc.* **2025**, *147*, 36053.
- R. Daniel Little and co-workers: *J. Org. Chem.* **2013**, *78*, 2104.
- Phil Baran and co-workers: *J. Am. Chem. Soc.* **2017**, *139*, 7448.
- R. Daniel Little: *J. Org. Chem.* **2020**, *85*, 13375 (perspective).

In fact, we have demonstrated that ferrocene, which exhibits a cyclic voltammogram similar to that of intermediate **V**, acts as an efficient electrocatalyst in the trifluoromethylation of substrate **1**, despite the similar uphill oxidation in solution.

Moreover, we have investigated the relative efficiencies of the different Fe species in solution in promoting photoinduced LMCT decarboxylation to generate CF_3 radicals by monitoring the photoreduction of in situ-formed ligated and non-ligated Fe(III) trifluoroacetate species, corresponding to intermediates **IV** and **V**, respectively. To this end, mixtures of $\text{Fe}(\text{OTf})_3$, NaO_2CCF_3 and TFA in acetonitrile were prepared in the presence and absence of **L1**, and their evolution under irradiation was followed by UV-Vis absorption spectroscopy.

Photodecarboxylation of non-ligated Fe(III) trifluoroacetate species (absence of **L1**)

(a) UV-Vis spectra of a mixture of $\text{Fe}(\text{OTf})_3$ (10 mM), NaO_2CCF_3 (600 mM) and TFA (130 mM) in dry acetonitrile, at different irradiation times (irradiation at 405 nm) [orange gradient]; (b) UV-Vis spectrum of a mixture of $\text{Fe}(\text{OTf})_2$ (10 mM), NaO_2CCF_3 (600 mM) and TFA (130 mM) [purple line].

After 30 minutes of irradiation, the non-ligated Fe(III) trifluoroacetate species associated with intermediate **IV** was completely consumed, as evidenced by the disappearance of the absorption band at 350 nm. Concomitantly, Fe(II) trifluoroacetate species were formed, as confirmed by comparison with the purple reference spectrum. The photoreduction of Fe(III) to Fe(II) arises from the decarboxylative LMCT process.

Photodecarboxylation of ligated Fe(III) trifluoroacetate species (with L1)

(a) UV-Vis spectra of a mixture of Fe(OTf)₃ (10 mM), **L1** (10 mM), NaO₂CCF₃ (600 mM) and TFA (130 mM) in dry acetonitrile, at different irradiation times (irradiation at 405 nm) [*orange gradient*]; (b) UV-Vis spectrum of a mixture of Fe(OTf)₂ (10 mM), **L1** (10 mM), NaO₂CCF₃ (600 mM) and TFA (130 mM) [*purple line*].

Photoreduction of the ligated Fe(III) trifluoroacetate species is evidenced by the emergence of an absorption band in the 500–600 nm region, which is attributed to Fe(II) intermediates based on comparison with the purple reference spectrum.

Photoreduction of Fe(III) species in the presence and in the absence of ligand **L1**. The inset shows zoomed data up to 0.5 h, indicating the photoreduction of both Fe(III) species after 30 min.

Notably, in the presence of **L1** the LMCT photodecarboxylation proceeds much more slowly than for the corresponding non-ligated species, requiring approximately 20 h to reach completion. Taken together, these results indicate that the photodecarboxylation of trifluoroacetates proceeds much more rapidly from non-ligated Fe(III) species (intermediate **IV**). This observation suggests a plausible shift in the redox equilibrium that overcomes the 200 mV uphill difference detected in our system, thereby providing support for pathway B of the proposed mechanism.

Finally, we have been also able to detect non-ligated Fe trifluoroacetate species in solution by in-situ HRMS monitorization, providing further support to the equilibrium displayed in Fig. 2 in the manuscript, resulting from a 1:1 mixture of Fe(II) and **L1** in the presence of trifluoroacetates and TFA.

Compound Details

Cpd. 1: C6 F9 Fe O6

Name	Formula	RT	RI	Mass Diff (Tgt, ppm)	CAS	ID Source	Score	Algorithm
	C6 F9 Fe O6	0.258		392.8947		FBF	96.98	FBF

Species	m/z	Score (Tgt)	Score (Lib)	Score (DB)	Score (MFG)	Score (RT)
M-	392.8951	96.98				

Compound Spectra (overlaid)

Spectrum Peaks

m/z	Z	Abund	Diff (ppm)	Height %	Height % (Calc)	Ion Species	Formula
392.8951	1	2294	-0.56	7.19	6.36	M-	
393.9001	1	286	3.58	0.89	0.43	M-	
394.8906	1	31932	0.02	100.00	100.00	M-	
395.8934	1	2968	0.28	9.30	9.02	M-	
396.8941	1	880	-0.06	2.76	1.88	M-	
397.8990	1	59	5.51	0.18	0.14	M-	

Compound ID Table

Name	Formula	Species	RT	RT Diff	Mass	CAS	ID Source	Score	Score (Lib)	Score (Tgt)
	C6 F9 Fe O6	M-	0.258		392.8947		FBF	96.98		96.98

(2) Using the Nernst equation to predict how much IV (A⁺) will be formed ... if an equivalent amount of V (Ox) is added to I (A), and II is (Red), the following equation describes the reaction equilibrium:

The ratio of [A⁺] to [A] is at equilibrium is given by:

$$\text{Log}\left\{\frac{[A^+]}{[A]}\right\} = 8.47\Delta E$$

where ΔE is $E(Ox/Red) - E(A^+/A)$. Thus, in this case, ΔE is $-0.2V$, and there is subsequently approximately $50\times$ more A in solution than A⁺. Given this unfavorable equilibrium implied in Path B, why do the authors maintain its importance in the overall reaction mechanism? Of course, this Nernstian treatment assumes that both the V/II and IV/I redox couples are reversible, and the latter is not. Thus, this unfavorable ET could be driven by downstream irreversible chemistry involving IV, such as LMCT. But why would pathway B, in which thermodynamically unfavorable oxidation is required, be competitive with Path A? LMCT

quantum yields are also notoriously low, meaning this second irreversible step is not facile. The authors state in their rebuttal that “although intermediate V is photoactive, its evolution might be slow enough to compete with a fast electron transfer with intermediate I in solution.” Is evolution of V slow (no electrokinetics are provided), or is electron transfer to I fast (once again no kinetics). Thus this statement regarding the viability of Path B is purely speculative. V is most likely photoactive and thus can be generated at the electrode, and therefore pathway A can account for all reactivity.

We thank the reviewer for the thorough evaluation of our work. While a Nernstian treatment may offer a general overview of a redox equilibrium, we believe it does not accurately represent the situation under our reaction conditions. As the reviewer noted, the Nernst equation assumes equilibrium to determine the relative concentrations of intermediates. However, the redox pair **I/IV** exhibits an irreversible CV, which challenges this assumption. Moreover, the Nernst treatment imposes equal concentrations of species **V** and **I**, a scenario that might not reflect the actual conditions in our system and would be speculative at this point.

Nevertheless, we agree that the unfavorable electron transfer could be driven by the irreversible photoinduced LMCT decarboxylation process at intermediate **IV**. To examine this point more carefully, we have now experimentally demonstrated that the photoreduction of **IV** (without **L1**) is complete within 30 min, whereas ligated species **V** requires approximately 20 h (see experimental details above). This experiment indicates that the photodecomposition of **IV** can be sufficiently rapid to drive, and effectively drain, the otherwise slightly unfavorable (about 200 mV) oxidation of **I** by **V** in solution. This is a similar behavior observed in other mediated redox processes reported in the literature (see above).

In our view, these observations further support path B as a viable and competent mechanistic pathway under our optimized reaction conditions.

(3) The authors state in their rebuttal that experimental evidence for Path B is that without L1, no product is generated. The absence of L1 would also shut down path A, which more than likely explains the lack of reactivity. The authors also show the ability for redox mediators to restore reactivity, but this is not relevant under catalytic conditions when L1 is present. It is even more irrelevant when the new ligand loading experiments are included, which

conclusively show product yield even with 50% ligand loading, conditions where ligandless FeTFA is expected to be practically nonexistent. Interestingly, it appears the authors get better results when using ferrocene instead of L1 (see Fig 3). Why was this not pursued further? Ferrocene is actually cheaper than L1 as well (by a factor of 40 based off Sigma and Ambeed prices), why not use it to mediate the transformation instead of L1 when the economics and yield are strictly better?

We agree with the reviewer that when using 50 mol% of **L1**, path A becomes more plausible. One advantage of the present protocol is that both pathways (A and B) can operate to deliver the desired products under optimized reaction conditions, when only 10 mol% of **L1** is used.

Regarding the use of ferrocene as a redox mediator in the absence of **L1**, while it yielded good results for the trifluoromethylation of substrate **1**, its efficiency varies with other substrates. This variation might be attributed to a competitive trifluoromethylation of ferrocene, a phenomenon we have observed in our previous trifluoromethylation work, which could reduce its overall efficiency. In fact, when ferrocene was used as a redox mediator in the trifluoromethylation of caffeine in the absence of **L1**, no product formation was observed. Presumably, for less electron-rich substrates such as caffeine, the homoleptic tris-bipyridine-type Fe(III) intermediate (**VI**) could participate in mediating rearomatization in the bulk solution, a process consistent with the anodic potential vs. yield relationship observed for product **19**, which plateaued around the redox pair **III/VI** (Fig. 6A).

In summary, in light of the above comments, I believe Path B should be discarded unless the reaction is run without L1 and with redox mediators. It seems likely that, under catalytic conditions, only Path A is operative. And this path is no different than standard electrophotocatalytic protocols. Path B may be operative with no ligand included, but then obviously Path A would not be functioning. The title is “Dual-Role Iron species...” but Fe would only be playing a dual role if Paths A and B were operative at the same time, and this is not necessary for product formation and also is not proven to be the case. The proposed dual-role mechanism is overcomplicated for novelty’s sake, and can be reduced to simply one integrated electrophotochemical cycle (Path A).

Unfortunately, this does decrease one of the main selling points of the paper. My original recommendation was publication if the proposed reaction mechanism could be supported and

if the methodology could be differentiated from prior work. Due to the “dual-role” mechanism not being in play anymore, as well as the scope not being able to be expanded beyond that of the authors’ own prior work (comment #10), I leave it up to the Editor to decide if this paper should be published. Regardless, the current version of the paper should not be published unless they provide definitive proof that Pathway B is operative.

We firmly believe that in this second revision of our work, we have provided additional evidence that strongly supports path B of our proposed mechanism, further clarifying the "dual role" of the in situ-generated iron species, a point that has been appreciated by reviewers #1 and #2.

Overall, we greatly appreciate the reviewers' comments, as their feedback throughout the revisions has encouraged us to investigate the mechanism in much greater detail.